# Local electronic descriptors for solute-defect interactions in bcc refractory metals

Yong-Jie Hu [1], Ge Zhao[2], Baiyu Zhang[3], Chaoming Yang[1], Mingfei Zhang [1], Zi-Kui Liu [4], Xiaofeng Qian[3] & Liang Qi [1]*

The interactions between solute atoms and crystalline defects such as vacancies, dislocations, and grain boundaries are essential in determining alloy properties. Here we present a general linear correlation between two descriptors of local electronic structures and the solute-defect interaction energies in binary alloys of body-centered-cubic (bcc) refractory metals (such as W and Ta) with transition-metal substitutional solutes. One electronic descriptor is the bimodality of the $d$-orbital local density of states for a matrix atom at the substitutional site, and the other is related to the hybridization strength between the valance $sp$- and $d$-bands for the same matrix atom. For a particular pair of solute-matrix elements, this linear correlation is valid independent of types of defects and the locations of substitutional sites. These results provide the possibility to apply local electronic descriptors for quantitative and efficient predictions on the solute-defect interactions and defect properties in alloys.

[1] Department of Materials Science and Engineering, University of Michigan, Ann Arbor, MI 48109, USA. [2] Department of Statistics, Pennsylvania State University, State College, PA 16802, USA. [3] Department of Materials Science and Engineering, Texas A&M University, College Station, TX 77843, USA. [4] Department of Materials Science and Engineering, Pennsylvania State University, State College, PA 16802, USA. *email: qiliang@umich.edu

Solute atoms, whether they are added voluntarily for specific needs, inevitably remained as impurities after the synthesis, or introduced during the materials service, can affect various properties of alloys by changing the stability and mobility of crystalline defects[1–5]. One characteristic example is body-centered-cubic (bcc) refractory alloys based on group V (V, Nb, Ta) and VI (Mo, W) elements. These alloys are usually composed of a single bcc solid–solution phase, of which many properties are mainly managed by controlling the interactions of crystalline defects with solute elements, especially transition metal elements[4,6–10]. These interactions can be quantitatively characterized as the solute–defect binding energy, which is often correlated with the elastic strain energy variations caused by the size mismatch between solute and matrix atoms at different atomistic sites[11–13]. Beyond elastic interactions, especially in/near the core regions of defects, the variations in local electronic structures and chemical bonding caused by solute and defect geometries should contribute to the solute–defect binding energies, so this variation is usually referred to as the electronic contribution in the literature[14,15]. Understanding and quantifying these electronic contributions are critical for both fundamental science and technological development of advanced alloys in future.

Scientifically, a general physics-based model is required to explain electronic effects on the solute binding for various types of defects and alloys recently found by first-principles calculations. The solute–defect binding in bcc refractory metals seems to show strong dependences on the electronic features of solute elements. A unique regularity—the solute–defect interaction becomes more attractive when the solute element has more valence electrons—has been reported for the interactions between transition metal elements and various types of crystalline defects in W/Mo alloys in different dimensions, including vacancies[16], dislocations[4,6,17], and grain boundaries (GBs)[18].

Technically, quantifying the electronic contributions may provide effective and robust descriptors to represent the features of materials in the complex compositional and structural spaces. Both first-principles calculations and atomistic simulations using empirical potentials are often difficult to provide computationally efficient and chemically accurate descriptions for various types of complex defects simultaneously, especially for alloy systems. The recent development of data-centric materials science based on machine learning methods may help resolve the problem. However, these new methods usually require the descriptors derived from physical principles to improve their transferability[19–21]. Electronic structures related to defect–solute interactions can be potential candidates for such descriptors, which have been suggested by many recent first-principles calculations. Some of these studies were related to electronic band filling effects[14,22,23]; others also indicated alternative electronic structure features that can affect energetic properties of the transition metal alloys, including $d$-band bimodality[24], the transition between $e_g$ and $t_{2g}$ orbital sets[25], $e_g/t_{2g}$ population ratio[17], and upper band edge[26].

Using first-principles calculations based on density functional theory (DFT), herein we show that the binding behavior between transition metal substitutional solute elements and various types of crystalline defects (zero-, one- and two-dimensional (0D, 1D, and 2D, respectively)) in non-magnetic bcc refractory metals is highly correlated to the variations in the local electronic structures of the matrix atom in the unalloyed defect. This correlation largely depends on two electronic descriptors inspired by tight-binding theory[24,27–30]. One descriptor is the variation in the bimodality feature of the $d$-orbital local density of states (LDOS) of the matrix atom before substitution; the other is the change in the bond hybridization strength between the valance $sp$- and $d$-bands of the same matrix atom. Moreover, based on these two electronic descriptors, a linear regression model is proposed to describe the solute–defect interaction energies in binary alloys of bcc refractory metals with transition metal substitutional solutes. For a particular pair of solute–matrix elements, this linear correlation is valid independent of types of defects and the locations of substitutional sites. We also provide detailed examples to demonstrate the promising potential of this correlation for efficient predictions of the defect–solute interaction energies at different atomic sites in complex defect structures. The prediction accuracy can be further improved by a residual-corrected nonparametric regression model solely based on descriptors established from the local electronic structures of the matrix atom. The observed generality of the solute–defect interaction can provide physical guidance on the proper selection of solute elements in a quantitative manner to control the crystalline defects in alloys with targeted properties.

## Results

**Solute interaction and LDOS of dislocation core.** Figure 1a shows the calculated interaction energy (i.e., binding energy) $E_{int}$ between the $\frac{1}{2}\langle 111 \rangle$ screw dislocation core and five types of transition metal substitutional solutes in bcc W, namely, Ta, Re, Os, Ir, and Pt. In this paper, positive/negative values of $E_{int}$ indicate attractive/repulsive interactions between solutes and defects. The dislocation structure is fully relaxed to reach its equilibrium state in pure W and subsequently used for solute substitution. The interaction energies are calculated under two conditions: relaxing and fixing atomic positions during the total energy calculations of the solute-doped dislocation structures. Therefore, the difference between the relaxed $\left(E_{int}^{relax}\right)$ and fixed-lattice interaction energies $\left(E_{int}^{fix}\right)$ gives the energy gained by the relaxation of the W lattice upon the solute substitution. As shown in Fig. 1a, both the relaxed and fixed-lattice interaction energies are negative for the solute with fewer $d$ electrons than W and become more positive when the solute has more $d$ electrons. In addition, the relative difference between $E_{int}^{relax}$ and $E_{int}^{fix}$ is small for all the solutes. These results indicate that the observed dependence of the interaction energies on the number of $d$ electrons of the solute element mainly originates from the local changes in the electronic structure near the dislocation core rather than the effects of the lattice relaxation upon the solute substitution.

Owing to the localized characteristics of $d$ orbitals, the LDOS of transition metals can display considerable shape features that are characteristic of the given crystal structure[27,29]. Using W as an example, Fig. 1b shows that the bcc structure results in a bimodal $d$-band LDOS (solid-blue line) with a pseudo-band gap in the middle of the $d$-band, while the LDOS of close-packed structures (i.e., face-centered cubic (fcc)/hexagonal close-packed (hcp)) has a unimodal shape (solid-orange line). Interestingly, it is found that the LDOS of the W atom surrounding the screw dislocation core (dashed-blue line) also has a less bimodal shape compared to that of perfect bcc, as a consequence of the change in local atomistic structures. Similar variation in LDOS is also observed for the $\frac{1}{2}\langle 111 \rangle$ screw dislocation in Nb and Mo[31]. The bimodality distinction of LDOS was found previously to be essential for differentiating the energetic stabilities between the bulk phases with bcc and close-packed structures in transition metal systems[27–29]. When $d$-band is about half-filled, the Fermi level ($E_F$) is located close to the minimum of the pseudo-band gap in the LDOS of bcc structure, as shown in Fig. 1b. Qualitatively speaking, the LDOS of bcc structure has more occupied states far below $E_F$ and less occupied states close to $E_F$ compared to that of fcc/hcp structure when the $d$-band is about half-filled[29]. This leads to a lower electronic band energy, which makes bcc structure more stable compared to the close-packed structure[29].

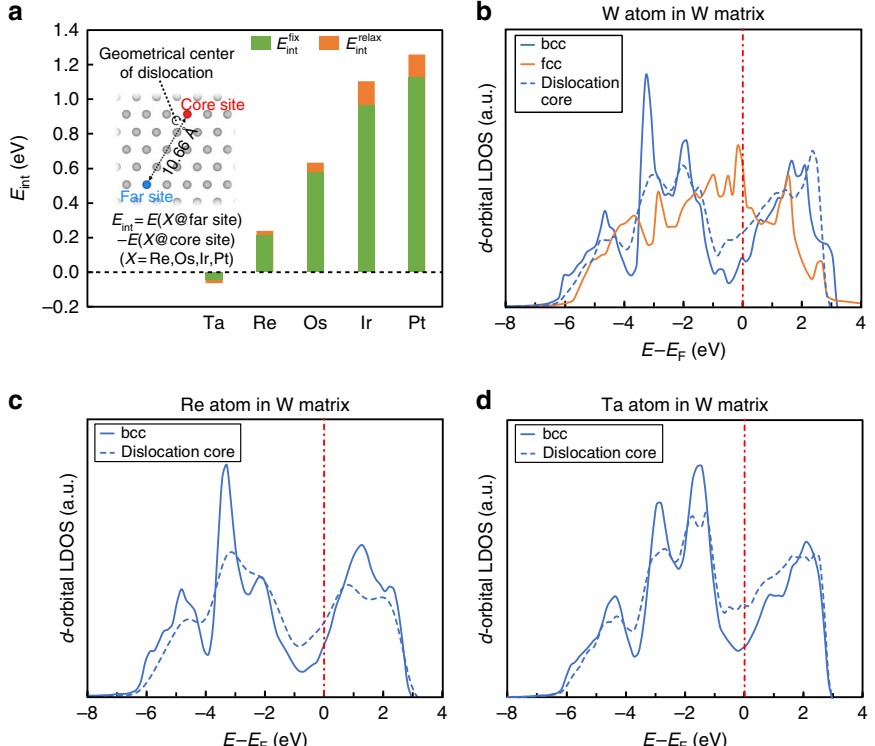

**Fig. 1** Solute-interaction energy and electronic structures of the $\frac{1}{2}\langle 111\rangle$ screw dislocation in bcc W. **a** Interaction energy between transition metal solute and the $\frac{1}{2}\langle 111\rangle$ screw dislocation in bcc W. The interaction energy, $E_{int}$, is defined as the difference between the total energies of the dislocation structure with a solute atom X occupying an atomic site far away from and at the dislocation core. The dislocation structure is initially fully relaxed to reach its equilibrium state in pure W and sequentially used for solute substitution. A positive value of $E_{int}$ indicates an attractive binding tendency. The interaction energies were calculated under two conditions: relaxing $\left(E_{int}^{relax}\right)$ and fixing atomic positions $\left(E_{int}^{fix}\right)$ during the total energy calculations after the solute atom added into the supercell. The values of $E_{int}^{relax}$ are taken from our recent publication[6]. **b** LDOS of a W atom in perfect bcc lattice (solid-blue line), perfect fcc lattice (solid-orange line), and at the $\frac{1}{2}\langle 111\rangle$ dislocation core site (dashed-blue line) in pure W. **c**, **d** LDOS of an Re and Ta atom occupying the bcc site (solid-blue line) and the $\frac{1}{2}\langle 111\rangle$ dislocation core site (dashed-blue line) in the W matrix, respectively. The bcc bulk site and core site refer to the atomic sites marked in blue and red colors in **a**, respectively

Interestingly, solute substitutions do not significantly change the bimodality features of LDOS for the dislocation core and the bcc bulk site, showing characteristics of the so-called canonical $d$-band[27,29,32]. Figure 1c, d show the LDOS of atoms at a dislocation core site and a bulk bcc site far away from the core when these sites are occupied by Re or Ta instead of W, respectively. The solute atom at the core site still has a less bimodal LDOS compared with its counterpart at the bulk site. However, the filling fraction of the local $d$-band of the solute atom is changed as it has a different number of $d$ electrons than W. As Re has more $d$ electrons than W, the position of the $E_F$ on LDOS of Re shifts away from the minimum of the pseudo-band gap, toward the right band edge. Moreover, it is found that $E_F$ will keep shifting closer to the right band edge for the solute with more $d$ electrons (Supplementary Fig. 6). According to bond-order potential theory, a structure with less bimodal DOS can usually be stabilized when the filling fraction is towards to the band edges, while a more bimodal DOS is favored for a half-filled band[27–30]. Therefore, compared to placing W atoms at the core site, the system may benefit from a stabilization contribution from the band energy when the core site is occupied by the solute atom with more $d$ electrons than W. Correspondingly, there is a positive/attractive interaction tendency between the dislocation core and these solute elements as shown in Fig. 1a. A similar solute-induced stabilization mechanism has also been demonstrated on the $\{112\bar{1}\}$ twin boundary (TB) of hcp Re[24]. On the other hand, compared to that of the W atom, $E_F$ shifts to a position even closer to the minimum of the pseudo-band gap of

the LDOS of the Ta solute as shown in Fig. 1d. Since the difference in the number of the occupied state close to $E_F$ between the core and bulk LDOS may be maximized at the minimum of the pseudo-band gap, Ta atom should be less preferred by the core site than W atom by considering occupied states close to and far below the $E_F$. This consequently yields a negative/repulsive interaction energy as shown in Fig. 1a.

**Electronic attributes of solute–defect interactions**. The results of Fig. 1 reveal a qualitative correlation between the $d$-band bimodality and the solute–dislocation interaction in the binary alloys of bcc W and transition metal solutes. To further explore this correlation, we investigate the local electronic structures of atoms near several 0D, 1D, and 2D defects in pure W, including mono-vacancy, <100>-dumbbell, <111>-dumbbell, $\frac{1}{2}\langle 111\rangle$ screw dislocation, $\Sigma 3(11\bar{2})$ TB, $\Sigma 3(111)$, $\Sigma 5(310)$, and $\Sigma 5(210)$ GBs. To quantify the bimodality of the DFT-calculated LDOS, Hartigan's dip test was performed[33,34]. A completed unimodal LDOS corresponds to a test statistic of 0, while a more bimodal LDOS has a larger value of test statistic[33,34]. We then use a parameter, $\Delta$dip, to quantify the change in the bimodality of the LDOS of the atoms near the defect relative to a reference atom that is far away from the defect, where $\Delta$dip = dip(reference) − dip(defect). Therefore, W atom at a site with a more positive $\Delta$dip will have a less bimodal LDOS compared to the atom at the reference site. Furthermore, for the W atoms where the $\Delta$dip calculations are performed, we also calculate the corresponding

fixed-lattice solute–defect interaction energies $(E_{int}^{fix})$ when these W atoms are substituted by the Pt, Re, and Ta solutes, respectively. The results are summarized in Supplementary Note 2. In addition, like the solute–dislocation interactions, it is found that the effects of solute-induced lattice relaxation on the interaction energy are also small for other defect structures in W (details in Supplementary Note 3).

By comparing the calculated $\Delta$dip with $E_{int}^{fix}$, we notice a very interesting phenomenon that the variations in $E_{int}^{fix}$ of the Re and Pt solutes are strongly correlated with the variations in the bimodality of the LDOS for the W atoms that is being substituted at the sites with different separation distance to the defect center. Taking the $\frac{1}{2}\langle 111\rangle$ screw dislocation as an example, as shown in Fig. 2a, the defect site with a higher $\Delta$dip generally has a more attractive interaction with the solutes (higher $E_{int}^{fix}$). This correlation is consistent with the analyses in Fig. 1b–d, since a more positive $\Delta$dip corresponds to a less bimodal LDOS feature for W atom at that site. If we assume that the solute substitutions do not significantly change the bimodality features of LDOS as shown in Fig. 1c, d, a less bimodal LDOS indicates that this atomic site prefers to be occupied by the solute atoms with more $d$ electrons than W because $E_F$ will be at a position closer to the edge of their $d$-band. In addition, the correlation between $\Delta$dip and $E_{int}^{fix}$ is found to be also valid for the Re and Pt solutes interacting with the defects in transition states, such as the generalized stacking faults (GSF) shown in Supplementary Note 4.

Moreover, if we plot all the calculated $E_{int}^{fix}$ together with respect to the corresponding $\Delta$dip parameter, an approximately linear relationship can be revealed between $E_{int}^{fix}$ and $\Delta$dip for both Re- and Pt-substitutional solutes, as shown in Supplementary Fig. 11a, b, respectively. These results indicate that the filling energy of the $d$-band associated with the bimodality variation indeed has significant contribution to the solute–defect interaction energy, which can be quantitatively described by the $\Delta$dip parameter. On the other hand, compared to the W–Re and W–Pt systems, the correlation between $E_{int}^{fix}$ and $\Delta$dip in the W–Ta system becomes more scattered. For example, as shown in Fig. 2b, the Ta solute generally interacts in a repulsive way with the W $\Sigma3(11\bar{2})$ TB, which yields a negative correlation between $E_{int}^{fix}$ and $\Delta$dip ($\Delta$dip $> 0 \rightarrow E_{int}^{fix} < 0$), consistent with the analyses in Fig. 1d. However, quantitative discrepancies can be seen for several individual sites near the defects. For example, sites 4 and 5 in $\Sigma3(11\bar{2})$ TB shown in Fig. 2b have nearly zero values of $\Delta$dip and notable values of $E_{int}^{fix}$ in contrast. This implies that there could be other underlying mechanisms contributing to the solute–defect interaction energies, which cannot be solely described by the $\Delta$dip term.

One possible mechanism could be the energy contributions from the valence $sp$-band. Owing to the covalent feature of the $d$-band, the valence $sp$-band can be strongly hybridized with and thus strongly influenced by the valence $d$-band. Within a tight-binding framework[35–42], the strength of the $sp$–$d$ hybridization $(E_{sp})$ of an atom in transition metal alloys can be correlated with a function of (i) the interatomic distances between the atom and its neighboring atoms $(d_{ij})$ and (ii) the spatial extents of the $d$-orbitals of the atom and its neighboring atoms $(r_{d_i} \& r_{d_j})$, which is $E_{sp} \propto \sum_j r_{d_i}^{\frac{3}{2}} r_{d_j}^{\frac{3}{2}}/d_{ij}^5$ (see Supplementary Note 5 for details). This suggests that the strength of the $sp$–$d$ hybridization in a defect structure should vary with each individual atom since $d_{ij}$ of the atom at each defect site can be different and the $r_{d_i}$ of the solute element can differ from that of the neighboring matrix element. Therefore, the effect of the $sp$–$d$ hybridization may not be ignored

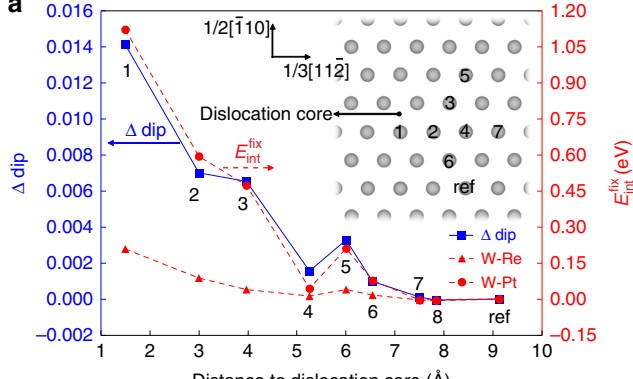

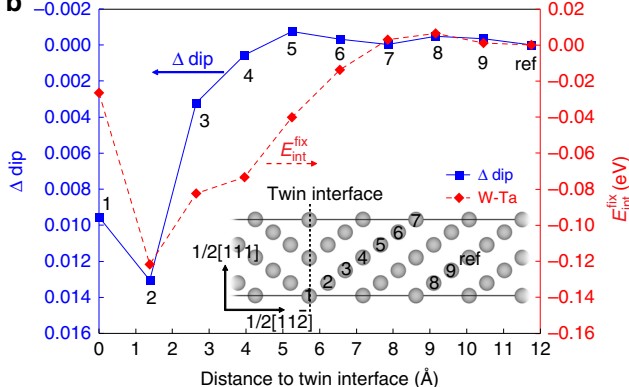

**Fig. 2** Correlation between $\Delta$dip of LDOS and $E_{int}^{fix}$ in W binary alloys. **a** $\frac{1}{2}\langle 111\rangle$ screw dislocation; **b** $\Sigma3(11\bar{2})$ twin boundary (TB). $E_{int}^{fix}$ refers to the interaction energy calculated based on the defect structures that are already fully relaxed in pure W and without further relaxing atomic positions after solute substitution. The calculated $\Delta$dip (blue squares) of each atomic site of interest in pure W, and the corresponding solute–defect interaction energy when the site is occupied by Re (red triangles), Pt (red circles), and Ta (red diamonds) are plotted with respect to the relative distance from the atomic site to the defect center. The positions of the atomic sites in the simulation cell are marked by numbers according to the pairing distance with the defect center and the investigated site. It should be noted that the axis value for the $\Delta$dip term in **b** is plotted in reverse order (i.e. higher $\Delta$dip values pointing downward)

for determining solute–defect interactions in the bcc refractory alloys.

**General correlation between electronic descriptors and $E_{int}^{fix}$.** Based on the discussion above, we propose a linear regression model that approximates the solute–defect interaction energy $(E_{int}^{fix})$ into two parts as shown in Eq. (1),

$$E_{int}^{fix} \approx \Delta E_d + \Delta E_{sp} \approx a_1\Delta\text{dip} + a_2 x_{sp} \qquad (1)$$

Here $\Delta E_d$ represents the energy contribution due to the $d$-band filling, which may linearly correlate with the changes in the bimodality of the $d$-band through the $\Delta$dip term and a fitting coefficient, $a_1$. The second part in Eq. (1), $\Delta E_{sp}$, represents the energy contribution related to the $sp$–$d$ hybridization. We propose that $\Delta E_{sp}$ can also be estimated through a fitting coefficient, $a_2$, and a variable, $x_{sp}$, that describe the local environment of the defect site related to the $sp$–$d$ hybridization.

In the present work, $x_{sp}$ of a matrix atom near the defect in pure metals is proposed to be,

$$x_{sp} = 1 - \frac{\left(V_{vor}^{def}\right)^{-\frac{5}{3}}/\epsilon_{sp}^{def}}{\left(V_{vor}^{ref}\right)^{-\frac{5}{3}}/\epsilon_{sp}^{ref}} \qquad (2)$$

where $V_{vor}^{def}/V_{vor}^{ref}$ is the Voronoi volume of the atom at the defect and reference site, respectively, and $\epsilon_{sp}^{def}/\epsilon_{sp}^{ref}$ is the center of the occupied $sp$-band projected on the atom at the defect and the reference site, respectively. The reference site is same as the one used for the calculation of $\Delta$dip and $E_{int}^{fix}$. The $\epsilon_{sp}^{def}$ term is calculated as

$$\epsilon_{sp}^{def} = \int_{-\infty}^{0} E\rho_{sp}^{def}(E)dE \Big/ \int_{-\infty}^{0} \rho_{sp}^{def}(E)dE \qquad (3)$$

where $\rho_{sp}^{def}(E)$ is the projected LDOS of the $sp$-band on the atom at the defect site and the Fermi energy $E_F$ is set to zero. $\epsilon_{sp}^{ref}$ is calculated in the same way for the atom at the reference site. In Eq. (2), Voronoi volume ($V_{vor}$) is used to describe the average changes in the interatomic distances ($d_{ij}$) of the atoms near the defect, and $1/\epsilon_{sp}$ is included as a scaling term to the effects of $sp$–$d$ hybridization on solute–defect interactions (see Supplementary Note 6 for details). Like the $\Delta$dip term, the Voronoi volume and LDOS of the $sp$-band are also determined from the DFT calculations of relaxed atomic structures of pure matrix metals that contain defects. Herein we expect that the electronic features of the matrix atoms at defects are mainly assessed by the $\Delta$dip and $x_{sp}$ parameters, while the fitting coefficient $a_1$ and $a_2$ should be fixed values for each matrix–solute element pair.

Based on Eq. (1), we perform linear regressions to model the DFT-calculated $E_{int}^{fix}$ of the crystalline defects in the W–Ta, W–Re, and W–Pt binary alloy systems. $\Delta$dip and $x_{sp}$ are treated as regression variables; $a_1$ and $a_2$ are fitting coefficients. As shown in Fig. 3, the solute–defect interaction energies ($E_{int}^{fix}$) predicted by the proposed linear model show good agreement with the results of DFT calculations for the W alloys with different transition metal solutes (i.e., Ta, Re, and Pt). Good regression quality is also demonstrated by the close-to-one value of adjusted $R^2$ as listed in Table 1.

Considering the closeness of the crystal and electronic structures between group V and VI bcc elements, one would naturally wonder whether Eq. (1) can also be generally applied to model the solute–defect interactions in the binary alloys of group V element and transition metal solutes. To explore the possible correlation, we also perform DFT calculations to calculate the $\Delta$dip and $x_{sp}$ of atoms in several 0D, 1D, and 2D crystalline defects in pure Ta. As expected, it is found that Ta atoms near the defect center also generally have a less bimodal LDOS compared to those far away. For example, the $d$-orbital LDOS for a Ta atom exactly on the interface plane of the $\Sigma3(11\bar{2})$ TB are plotted in Fig. 4a, showing less bimodal characteristics comparing to the LDOS of a Ta atom far away from the interface.

The fixed-lattice solute–defect interaction energies ($E_{int}^{fix}$) are also calculated correspondingly when Ta atoms are substituted by the Hf and Os solutes. Linear regressions based on Eq. (1) are performed to model the DFT-calculated $E_{int}^{fix}$. Parity plots of the regression results are shown in Fig 4b, c for Ta–Hf and Ta–Os systems, respectively. The regression coefficient and parameters are listed in Table 1. As shown by both Fig. 4 and Table 1, the proposed linear regression model (Eq. (1)) can be generally applied to quantitatively describe the solute–defect interactions in Ta-based alloys as well.

**Improving the accuracy of the linear correlation.** As shown in Figs. 3 and 4, a few of outliers still appear in the predictions of the linear regression model, which have apparent discrepancies from the DFT results. Interestingly, we found that these outliers usually repeatedly appear at particular defect sites in multiple alloying systems. Scrutinizing the local electronic structures of the matrix atom at these outlier sites, it is found that there are some additional local features in their LDOSs. These features could affect the solute–defect energetics but are not sufficiently described by the $\Delta$dip and $x_{sp}$ parameters, resulting in large prediction errors. More detailed explanation can be found in Supplementary Note 8.

The above finding suggests that the remaining residuals of the linear regression model can be reduced if the model includes some other descriptors of the electronic bands in addition to $\Delta$dip and $x_{sp}$. As indicated in the recent DFT calculations, the energetic properties of the transition metal alloys could connect closely with many band features, including the transition between $e_g$ and $t_{2g}$ orbital sets[25], $e_g/t_{2g}$ population ratio[17], band occupation fraction[14,22,23], and upper band edge[26]. Therefore, we propose an additional regression function, which is added on the basis of

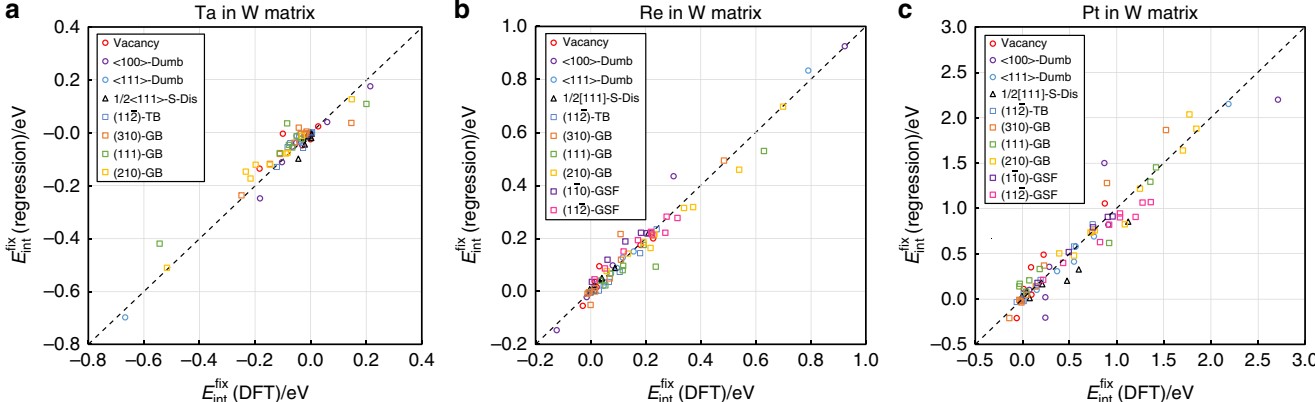

**Fig. 3** Comparison between the $E_{int}^{fix}$ from DFT calculations and predicted from Eq. (1) in the W binary alloys. **a** W–Ta system; **b** W–Re system; **c** W–Pt system. The data of the point, line, and planar defects are marked in circle, triangle, and square symbols, respectively. The DFT-calculated $E_{int}^{fix}$ refers to the solute–defect interaction energies calculated based on fixed atomistic structures that are already fully relaxed in pure W. In the legend, Dumb refers to the abbreviation of the dumbbell defects, S-Dis refers to the abbreviation of the $\frac{1}{2}\langle111\rangle$ screw dislocation, TB refers to twin boundaries, GB refers to grain boundaries, and GSF refers to generalized stacking faults. The values of the $\Delta$dip and $x_{sp}$ of each defect site used for the linear regression are listed in Supplementary Table 2. The regression parameters for each matrix–solute element pair are summarized in Table 1

Eq. (1) to further correct the remaining residuals from the linear regression. Accordingly, the solute–defect interaction energy $\left(E_{\text{int}}^{\text{fix}}\right)$ is now proposed to be approximated as,

$$E_{\text{int}}^{\text{fix}} \approx a_1 \Delta \text{dip} + a_2 x_{\text{sp}} + f_{\text{r-c}}\left(D_i, D_j, \dots\right) \qquad (4)$$

where the first two parts of the equation are the linear model described by Eq. (1) with the same $a_1/a_2$ from Table 1. $f_{\text{r-c}}(D_i, D_j, \dots)$ is the residual-correction function established by regressing the residuals $\Delta_{\text{linear}}$ ($\Delta_{\text{linear}} \equiv E_{\text{int}}^{\text{fix}} - (a_1 \Delta \text{dip} + \alpha_2 x_{\text{sp}})$) of the linear model based on a boarder set of 23 potential electronic descriptors ($D_i$, $D_j$…). These descriptors include $\Delta \text{dip}$ and $x_{\text{sp}}$; they also contain the band center and root-mean-square width of the whole d-orbital, $e_g$ and $t_{2g}$ orbital sets, and the sp-orbitals. In addition, these descriptors include the individual bimodalities of the $e_g$ and $t_{2g}$ orbital sets. All of these 23 descriptors are available from the DFT calculations of the defects relaxed in pure metals of matrix elements. A detailed description of the descriptor construction is included in Supplementary Note 9.

In the present work, the residual-correction function, $f_{\text{r-c}}(D_i, D_j, \dots)$, is developed based on a sophisticated local regression model, as implemented in the Locfit package[43–46]. The model performs a series of kernel-weighted local linear regressions within a moving window across the descriptor space, which gives the largest weight to observations close to the center of the window and produces a smooth curve that runs through the middle of the observations[44–46]. The local regression is performed with only 4 of the 23 potential electronic descriptors at a time to mitigate the risk of overfitting. Within a cross-validation framework, we select five sets of descriptors (each set containing four descriptors) that provide the best regression accuracy on average in all the five solute–matrix systems studied in the present work, and all of these five descriptor sets have two or three descriptors in common. We then establish the residual-correction function by averaging the corresponding local regression models of these five sets of descriptors. More details on the algorithms and calculation procedures of this statistical model can be found in Supplementary Note 9.

The regression results of the improved model based on Eq. (4) (referred as the linear $+ f_{\text{r-c}}$ model in the following) are plotted against the original DFT data in Fig. 5a, b for the W–Re and Ta–Hf systems, respectively. The regression results from the linear model solely based on $\Delta \text{dip}$ and $x_{\text{sp}}$ (Eq.(1)) are also included for comparison. As shown in both figures, the developed linear $+ f_{\text{r-c}}$ model indeed yields better agreements with the original DFT results. The parity plots of the W–Ta, W–Pt, and Ta–Os systems are shown in Supplementary Fig. 17, where the improvement of the regression accuracy is also clearly observed.

**Prediction of solute segregation in complex GB structures.** Since all the descriptors used in the present linear correlation model and the regression model are available from the LDOSs of atoms at/near the relaxed defect structures in pure metals, one could possibly apply the model to efficiently predict the solute–defect interaction energy of any atomic sites in the defects of interest, especially those with complex geometries. Here we show some examples in both Ta and W matrix in terms of two complex GBs, namely the Σ13 (230) and Σ27 (552) GBs. These two GB structures both have high index GB planes and complex geometries, which require large supercells to accommodate (Supplementary Fig. 4). Particularly, the input geometry of the Σ27 (552)-GB is implemented from a ground state structure in W predicted by a state-of-art evolutionary structure search algorithm[47,48]. The prediction results of the linear (Eq. (1)) and the linear $+ f_{\text{r-c}}$ (Eq. (4)) model based on electronic descriptors

**Table 1 Coefficients and accuracies of the linear regression model**

| Alloy system | Coefficient | | Adjusted $R^2$ | RMSE |
|---|---|---|---|---|
| | $a_1$ | $a_2$ | | |
| W-Ta | −7.20 | 1.78 | 0.9260 | 0.043 |
| W-Re | 15.97 | −1.29 | 0.9614 | 0.038 |
| W-Pt | 61.58 | −1.02 | 0.9415 | 0.174 |
| Ta-Hf | 6.70 | 2.09 | 0.9351 | 0.042 |
| Ta-Os | −6.48 | −4.08 | 0.8840 | 0.106 |

The model is based on Eq. (1) with coefficients $a_1$ and $a_2$ for different matrix–solute element pairs. The accuracy of the model is evaluated by its adjusted $R^2$, which represents the proportion of the variance in the regression response that is predictable from the regression variables, and the root-mean-square-error (RMSE). The units of $a_1$, $a_2$, and RMSE are all eV

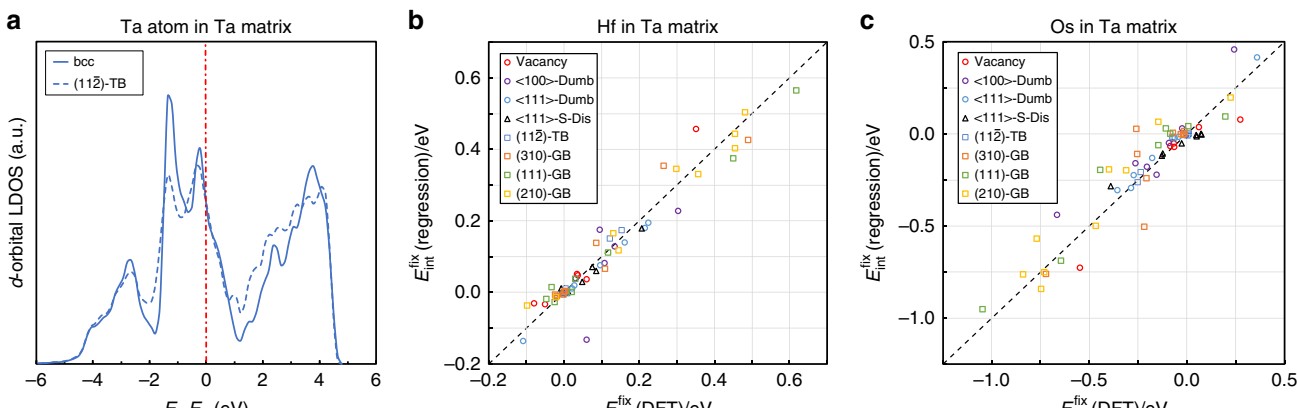

**Fig. 4** d-band bimodality and solute–defect interaction energies in bcc Ta. **a** Projected LDOSs of d orbitals of a Ta atom on the interface of the Σ3(11$\bar{2}$) TB (dashed line) and in bulk lattice (solid line), respectively. **b, c** DFT-calculated $E_{\text{int}}^{\text{fix}}$ in comparison with the predictions from the linear regression model in the cases of the Ta–Hf and Ta–Os systems, respectively. The DFT-calculated $E_{\text{int}}^{\text{fix}}$ refers to the solute–defect interaction energies calculated based on fixed atomistic structures that are already fully relaxed in pure Ta. In the legend, Dumb refers to the abbreviation of the dumbbell defects, S-Dis refers to the abbreviation of the $\frac{1}{2}\langle 111\rangle$ screw dislocation, TB refers to twin boundaries, and GB refers to grain boundaries. The values of $\Delta \text{dip}$ and $x_{\text{sp}}$ of each defect site used for the linear regression are listed in Supplementary Table 3. The regression parameters for each matrix–solute element pair are summarized in Table 1

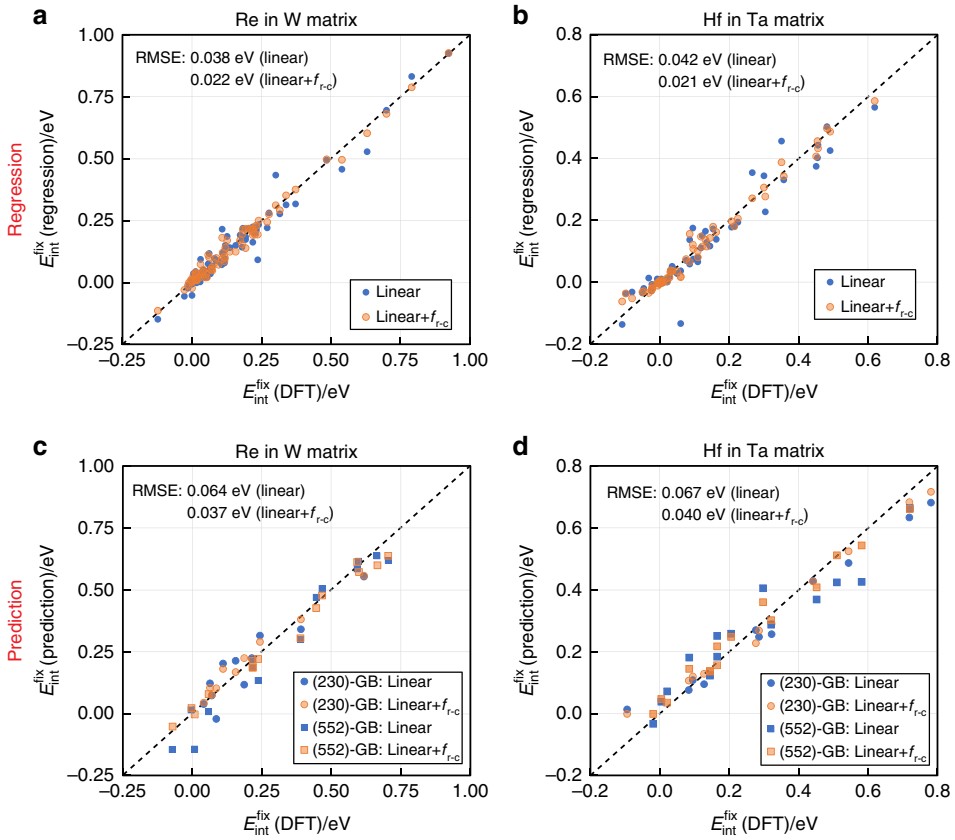

**Fig. 5** Comparison between the $E_{int}^{fix}$ from DFT calculations and from Eq. (1) (blue) and Eq. (4) (orange). **a**, **c** W–Re system; **b**, **d** Ta–Hf system. **a**, **b** Regression results based on the dataset in Figs. 3 and 4. **c**, **d** Prediction results of the Σ13 (230) GB and Σ27 (552) GB. The DFT data in **c**, **d** are not included in the regression process

from DFT calculations of the unalloyed GBs are shown as parity plots in Fig 5c, d for the W–Re and Ta–Hf systems, respectively, in comparison with the DFT-computed $E_{int}^{fix}$. As shown by the blue symbols, the predictions solely from the two-descriptor linear model have already reached fairly good agreements with the DFT results for both GBs in both systems, indicating that the major energy contributions to $E_{int}^{fix}$ can be well captured by the linear model alone. Moreover, by adding the residual-correction function ($f_{r-c}$), the linear + $f_{r-c}$ model (orange symbols) yields even better agreements, especially for the sites where the predictions of the linear model have large deviations. Similar validation results are also observed for the W–Ta, W–Pt, and Ta–Os systems, as shown in Supplementary Fig. 17.

With the predicted solute–defect interaction energies at each defect site, one can use the White–Coghlan site occupation model[49,50] to estimate the GB solute concentration isotherms under an assumption of non-interacting solutes,

$$c_{GB} = \frac{1}{N} \sum_{i=1}^{N} \frac{1}{1 + \frac{1-c_{bulk}}{c_{bulk}} \exp\left(-\frac{E_{int}^{X,i}}{k_B T}\right)} \quad (5)$$

where $E_{int}^{X,i}$ is the interaction energy of solute, X, when it occupies the $i$th of $N$ sites at GB, $T$ is temperature, and $c_{bulk}$ is the solute concentration in the bulk matrix (fixed as 2 at.% here). The solute concentration isotherms calculated using the $E_{int}^{X,i}$ predicted by both the linear and linear + $f_{r-c}$ model are compared with those calculated using DFT-computed $E_{int}^{X,i}$. As shown in Fig. 6a, b, for both of the GBs and all the five studied solute–matrix systems, the interaction energies predicted by the linear + $f_{r-c}$ model give

concentration isotherms that are very close to the DFT reference curves across a wide temperature range. The largest deviation is seen for the case of Pt in W (552)-GB at high temperature range at about 6 at.%. In fact, the curves calculated using the interaction energies solely predicted by the linear model are already in fairly good agreement with the DFT references, except for the case of Pt in W (552)-GB at low temperature.

These results suggest that, with the present model, one can estimate the interaction energies in complex defect structures with reasonably small uncertainty for the prediction of solute segregation isotherms. Instead of running many case-by-case calculations for substitutional solutes at different atomic sites surrounding a specific defect, only one DFT calculation for this defect in pure matrix metal is needed for obtaining the local electronic descriptors. Here it has to be emphasized that, although the root-mean-squared errors are 0.03–0.1 eV for defect–solute interaction energies (varying from ~−1.0 eV to ~+3.0 eV) for individual defect sites in these five matrix–solute pairs, we still obtain the reasonably good accuracy in the prediction of solute segregation because the concentration values depend on the defect–solute binding energies of multiple sites at/near the defects. There could be risk having large errors if the current linear or linear + $f_{r-c}$ model is applied to predict solute effects on defect properties that are sensitive to the solute interaction with a particular defect site.

## Discussion

There are two major aspects that require further investigations to understand and improve our proposed numerical model for

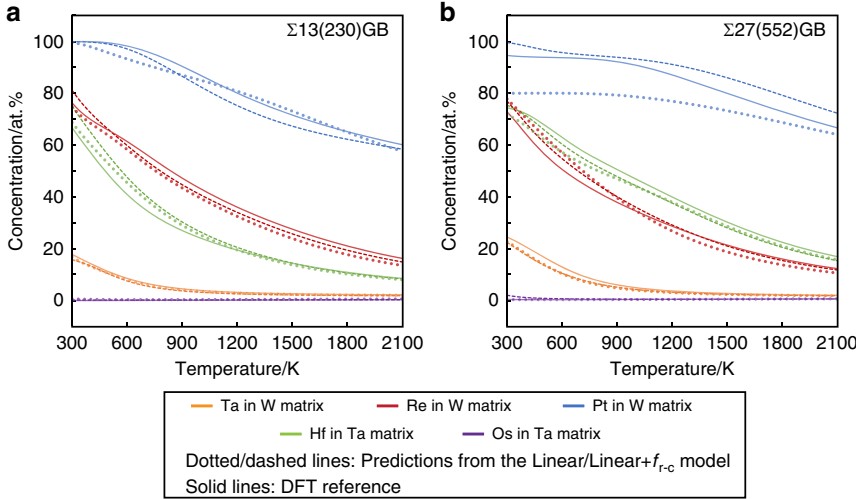

**Fig. 6** GB solute concentration isotherms calculated based on the WC model (Eq. (5)). **a** Σ13 (230)-GB; **b** Σ27 (552)-GB. The solid lines represent the isotherms calculated using DFT-computed interaction energies, while the dotted (dashed) lines represent the isotherms calculated using the interaction energies predicted from the linear (linear + $f_{r-c}$) model based on Eq. (1) (Eq. (4))

solute–defect interactions and defect properties in more general cases. For the first aspect, fundamental and quantitative physical mechanisms are needed to interpret the most effective descriptors and corresponding coefficients. As the linear correlation model is inspired by the moment analysis of DOS based on tight-binding theory[27–30], it would deepen our understanding of solute–defect interactions if we can also provide physical interpretation of the fitting coefficients.

The fitting coefficients ($a_1$ and $a_2$) in Table 1 indeed show strong dependence on the number of $d$ electrons of the solute element. In W alloys, the Δdip term yields a positive contribution ($a_1 > 0$) to $E_{int}^{fix}$ for the solute with more $d$ electrons than W (e.g., Re and Pt), while it yields a negative contribution ($a_1 < 0$) for the solute with fewer $d$ electrons (e.g., Ta), which is consistent with our analysis in Fig. 1b–d. In Ta alloys, this contribution becomes positive (negative) for the solute with fewer (more) $d$ electrons than Ta, e.g., Hf vs. Os. This is because the relative position of $E_F$ on the LDOS of the $d$-band is intrinsically different between Ta and W when they serve as the matrix element. As shown in Fig. 4a, $E_F$ of the Ta matrix is located on the lower energy side of the bcc pseudo-band gap, unlike the position of $E_F$ in the W matrix shown in Fig. 1b. Therefore, when alloying Ta and solutes with fewer (more) $d$ electrons, such as Hf (Os), the position of $E_F$ on the local $d$-band of the solute atom would further shift away from (toward) the pseudo-band gap compared to that of Ta matrix atom, leading to a positive (negative) contribution to $E_{int}^{fix}$ in terms of the Δdip parameter. Moreover, by alloying Ta with the solute element having even more $d$ electrons (e.g., Au), $E_F$ should continuously move across the pseudo-band gap to the right edge of $d$-band to generate a positive contribution to $E_{int}^{fix}$. Consequently, the energy contributions of the Δdip term in the alloys of group V elements should have an overall parabolic relationship with the number of $d$ electrons of solute, which may be reflected in some cases of the solute–defect interactions (e.g., Supplementary Fig. 18. and ref. [51,52]). In addition, in both Ta- and W-based alloys, the coefficient of the $x_{sp}$ term ($a_2$) always has a positive sign if the solute element has less $d$ electrons than the matrix element (e.g., W–Ta and Ta–Hf), while yields a negative sign if the difference in the number of $d$ electrons is reversed. This correlation can be understood in terms of the difference in the spatial extent of $d$-orbital between the solute and matrix elements. Details are provided in Supplementary Note 10. These qualitative

results provide the foundations for further investigations of physical mechanisms of solute–defect interactions in a quantitative manner in refractory metals and beyond.

For the second aspect, although the linear model could be robust for general solute–defect interactions since it is based on physics-inspired mechanisms, the residual-correction model should be further improved for more accurate and efficient prediction ability. As shown in Figs. 5 and 6, our current methods are reasonably accurate to predict the defect properties that depend on average effects of defect–solute interactions. However, improvements are still needed for predicting the individual defect–solute interaction at a specific defect site in the weak limit ($|E_{int}^{fix}| < \sim 0.05$ eV). Since the residual-correction functions were developed based on local regression method from the limited amount of data due to the large computational cost (351 regression data points for 5 matrix–solute element pairs), the natural strategy to improve the accuracy and transferability of our method is to include more solute–defect interactions data and apply more advanced regression methods.

Furthermore, more representative and deterministic descriptors of electronic and atomistic structures can further improve the accuracy of our method. The discussions in Supplementary Note 8 show that Δdip has limitations to describe the characteristics of $d$-band LDOS in specific situations. These problems are overcome by including other effective descriptors, such as the center of the $d$-band, the center of the $sp$-band, and Δdip of the $e_g$ orbitals, in the residual-correction model, but they may not be the final solutions. Moreover, the accuracy could be further increased if we apply certain descriptors from deterministic methods instead of Δdip, which have tiny fluctuations due to its statistical method associated with the random number generator. The fluctuations can cause prediction uncertainties on the level of $\sim 0.001$ eV. In addition, descriptors for atomistic structures can be included to consider the elastic contributions in the weak limit of interactions[13,53,54].

In summary, our findings establish a general and quantitative correlation between electronic structure descriptors and energetic stabilities of crystalline defects containing substitutional solute atoms in bcc refractory alloys. It is inspired by the classical theories of bulk phase stability based on electronic structures and applied to explain the energetic stabilities of local structural units at the atomistic level[24]. This correlation can potentially serve as a quantitative guideline for the transition metal alloy design with

targeted properties by controlling the effects of solute–defect interactions on defect stability and mobility. From a broader perspective, this study provides a robust example and a key step to construct advanced theories to describe the quantitative connections between the chemical bonding characteristics at the electronic level and the macroscopic materials' properties[55–57]. In addition, the observed electronic descriptors have potentials to be applied in data-centric materials' innovation based on machine learning techniques[58–60].

## Methods

**First-principles calculations.** First-principles calculations in the present work were carried out using the projector augmented wave (PAW)[61] method and the exchange-correlation functional depicted by the general gradient approximation from Perdew, Burke, and Ernzerhof[62], as implemented in the Vienna ab initio simulation package (VASP)[63]. The energy cutoff of the plane-wave basis was 400 eV. Brillouin zone integration was performed using a first-order Methfessel–Paxton smearing of 0.2 eV[64]. The grid of the $k$-point mesh in the first Brillouin zone is set according to the size and geometry of the simulation supercells (see Supplementary Method for details). The convergence criterion of the electronic self-consistent loop was set as $10^{-7}$ eV for the structure relaxation and $10^{-8}$ eV for the static calculations. The electronic configurations of the pseudopotentials used for the present first-principles calculations are summarized in Supplementary Table 1. As shown in Supplementary Table 1, the semi-core $5p$ electrons are treated as valence electrons for the calculations of Hf, Ta, and W. However, it is found that the LDOS of the $5p$-band localizes at very low energy states far away from the Fermi level and has a very large energy gap with the $5d$-, $6s$-, and $6p$-bands. We thus assume that the $5p$ electrons are basically inner-core electrons that have very limited contributions to electronic bonding. Therefore, the LDOS of the $5p$-band is not included in the band analysis based on Eq. (3).

First-principles calculations are performed in three steps to model the local electronic descriptors of the crystalline defects in bcc Ta and W and their interactions with substitutional solute atoms. In the first step, relaxation calculations are performed to obtain the optimized atomistic structures of crystalline defects in the pure metal matrix. In each relaxation calculation, the atoms and geometry of the simulation supercells are fully relaxed according to the Hellmann–Feynman forces, except calculations for the $\frac{1}{2}\langle111\rangle$ screw dislocation and the GSF defects due to their unique atomistic geometries. The relaxation of the $\frac{1}{2}\langle111\rangle$ screw dislocation is performed using the flexible boundary condition method[65,66]. The relaxation scheme consists of two steps: (1) the conjugate gradient relaxation of atoms near the dislocation core based on DFT calculations, and (2) the atomic structures outside the core region are relaxed based on the lattice Green function[4,6,65,66]. The two steps are repeatedly iterated until the maximum Hellmann–Feynman forces are <5 meV/Å[4,6]. In the calculations of the GSF defects, the atoms are only allowed to relax along the direction perpendicular to the fault plane. In the second step, static calculations are performed based on the relaxed defect structures to obtain the projected LDOS on each atom in the supercells. Then the local electronic descriptors of each atomic site of interest are obtained from the DFT-calculated LDOSs and atomistic structures. In the third step, solute atoms are introduced to substitute the individual solvent atoms with different separation distances to the defect center to investigate the solute–defect interactions. The relaxed defect structures in pure metals are used for solute substitution. After substitution, the interaction energies are then calculated under two different conditions: fixing and relaxing atomic positions during the total energy calculations of the solute-doped defect structures. The difference between the relaxed $\left(E_{\mathrm{int}}^{\mathrm{relax}}\right)$ and fixed-lattice interaction energies $\left(E_{\mathrm{int}}^{\mathrm{fix}}\right)$ gives the energy change due to the relaxation of the defect lattice upon the solute substitution. The fixed-lattice interaction energies are calculated for all solute–defect interactions considered in the present work, while the relaxed interaction energies are only calculated for a few defect sites in order to evaluate whether the lattice relaxation has a significant contribution to the solute–defect interaction energies. A detailed comparison between the calculated $E_{\mathrm{int}}^{\mathrm{fix}}$ and $E_{\mathrm{int}}^{\mathrm{relax}}$ is described in Supplementary Note 3.

**Hartigan's dip test.** The Hartigan's dip test is a statistical method proposed by Hartigan and Hartigan[34], which measures the deviation of the cumulative distribution function of an empirical distribution from that of unimodal distributions. The test takes a sample from the distribution density as inputs and transfers it into its unique corresponding cumulative distribution function, $F(x)$. Since the distribution is empirical, the corresponding $F(x)$ is a step function that jumps at each interval $\{x_i\}_{i=1}^n$, where $n$ equals to the number of total intervals. In the test, there are three major steps. First, based on all the possible intervals $[x_i, x_j]$ of $F(x)$, where $1 \leq i \leq j \leq n$, we generated a set of unimodal cumulative distributions function, $\left\{H_{ij}(x)\right\}_{1 \leq i \leq j \leq n}$, that are all close to $F(x)$. It means each of $H_{ij}(x)$ have to satisfy that: (i) the mode of $H_{ij}(x)$ is located in the interval $[x_i, x_j]$; (ii) $H_{ij}(x)$ is a straight line connecting $(x_i, F(x_i))$ and $(x_j, F(x_j))$; (iii) $H_{ij}(x)$ is the greatest one among all the convex functions that have smaller values than $F(x)$ in the range $(-\infty, x_i)$; and (iv) $H_{ij}(x)$ is the smallest one among all the convex functions that have larger values

than $F(x)$ in the range $(x_j, +\infty)$. Second, each of $H_{ij}(x)$ is vertically shifted upward and downward with a same distance, $d_{ij}$, to form a band. The shifting is stopped until $F(x)$ is within the band in all range, $(-\infty, +\infty)$. Then, this shifting distance, $d_{ij}$, is defined as the distance between $F(x)$ and $H_{ij}(x)$. Third, the smallest $d_{ij}$ among all the tested $H_{ij}(x)$ is defined as the dip test statistic, which is returned by the test. Therefore, the unimodal distribution corresponds to a statistic of 0, while a more significant bimodal distribution is evidenced by a larger statistic.

In the present work, to perform the Hartigan's dip test, the LDOS from first-principles calculations was normalized with respect to its total number of DOS and treated as an empirical distribution. The default settings in VASP was used to determine the minimum/maximum energy boundaries of the LDOS, so the interval of each individual LDOS calculation is slightly varied, ranging from 0.151 to 0.155 eV. Default setting was used for the NBANDS tag in the W-based calculations, which gave an average number of bands about 7.2 per atom. To keep the consistency, The NBANDS tag in the DFT calculations of the Ta system was set to the same value as those used in the W-based calculations. The sample for the dip test was then drawn randomly from the normalized LDOS with a size of 500 data points (Each LDOS in the present work was set to have 301 energy intervals in first-principles calculations.). We have drawn 8000 samples for each LDOS, and the dip test statistic of each LDOS being used for comparison is taken as the average of the statistics from the 8000 samples. All the Hartigan's dip tests of bimodality of LDOS were performed using a MATLAB code by Mechler[67]. In addition, the sensitivities of the Δdip measurements to the LDOS-related DFT parameters (i.e., the number of bins, $k$-point density, cutoff energy, and width of smearing) were tested, which is described in Supplementary Note 1. In addition, the performance of Eq. (1) on predicting the $E_{\mathrm{int}}^{\mathrm{fix}}$ calculated from the four-supercell method[68,69] are discussed in Supplementary Note 7.

## Data availability

The data that support the findings of this study are available from Supplementary Information and two public open-access repositories with identifiers (1) materials cloud (https://doi.org/10.24435/materialscloud:2019.0047/v1) and (2) materials commons (https://doi.org/10.13011/m3-k83c-kr76). The raw DFT data are also included in the open-access repositories.

## Code availability

The codes that support the findings of this study are available from the two public open-access repositories mentioned in the section of "Data availability."

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

## Acknowledgements

Y.-J.H., C.Y., M.Z., and Q.L. acknowledge the support by startup fund from the University of Michigan and the partial support by National Science Foundation (NSF) under award DMR-1847837. B.Z. and X.Q. acknowledge the startup fund from Texas A&M University and the partial support by the NSF under award number OAC-1835690. Z.-K.L. would like to acknowledge the partial financial support from the NSF grant CMMI-1825538. The calculations were performed by using the Extreme Science and Engineering Discovery Environment (XSEDE) Stampede2 at the TACC through allocation TG-DMR190035, the computational resources and services provided by Advanced Research Computing at the University of Michigan, Ann Arbor, the resources of the National Energy Research Scientific Computing Center, a DOE Office of Science User Facility supported by the Office of Science of the U.S. Department of Energy under Contract No. DE-AC02–05CH11231, and the advanced computing resources provided by Texas A&M High Performance Research Computing. Finally, we would like to thank Professor Dallas R. Trinkle in University of Illinois Urbana-Champaign for sharing his simulation codes on the flexible boundary condition method.

## Author contributions

Y.-J.H., X.Q. and L.Q. conceived the research and designed the modeling procedures. Y.-J.H., B.Z., C.Y. and M.Z. performed the first-principles calculations. Y.-J.H. and G.Z. performed the Hartigan's dip tests and the modeling of the residual-correction function. Y.-J.H., Z.-K.L., X.Q. and L.Q. prepared the manuscript. L.Q. supervised the project. All authors discussed the results and contributed to the manuscript.

## Competing interests

The authors declare no competing interests.
