## [Transparent Peer Review File · Nature Communications]

Reviewers' comments:

Reviewer #1 (Remarks to the Author):

The manuscript presents the results of DFT calculations of solute-defect interaction energies in body-centered cubic metals and proposes a correlation with the bimodality of the d-band local density of states close to the defect measured by the Hartigan's dip test through the Δ dip parameter. The present work mainly focusses on rhenium and platinum substitutional solutes interacting with crystalline defects in bcc tungsten (vacancy, dislocation core, twin boundary, grain boundary, generalized stacking fault). A linear relation between the unrelaxed solute-defect interaction energy and the Δ dip parameter is evidenced in tungsten. Based on these findings, supplemented by additional calculations of unrelaxed solute-GSF interaction energy in tungsten and tantalum for several 5d elements with a higher number of d electrons, a universal link between the Δ dip parameter and the unrelaxed interaction energy is claimed by the authors. Although this correlation is very interesting in Figure 4, from my point of view it is not sufficiently proved to be general and strong to be accepted to be published in Nature Communications.

This study is limited to substitutional solutes and this must be explicitly mentioned in the text. Meanwhile, the correlation is established for various defects in the case of unrelaxed interaction energies in W and the fact that the energy is unrelaxed has to be more explicitly written. To support the universality of their correlation, the authors must perform a more systematic set of calculations in at least one group V element (and not only generalized stacking faults in Ta) and for other 5d solutes with a lower number of d electrons in addition to the solutes with a higher number of d electrons than in W presented in the paper. Based on the qualitative energy band analysis of Figure 2 for the correlation with the variations in the bimodality of the LDOS, one would expect attractive binding energies in the case of i.e. Hf or Ta solutes in W.

Based on their calculations in tungsten, the authors assume that the effect of relaxation on the solute-defect interaction energies is small enough to be neglected further in the work. However, the authors might know that the group dependence of the relaxation around a defect can be important, and in particular it is well-established that relaxations are more important for group V elements than for group VI elements.

Some remarks:

- 1) The solute-GSF interaction energy has been calculated based on the commonly used definition: $E(\text{defect}+\text{solute})+E(\text{bulk})-E(\text{defect})-E(\text{solute})$. Why is this definition not used by the authors for the other defects instead of calculating the energy difference between the solute close to the defect and as far as possible from the defect?
- 2) Details about the pseudopotentials used in the present study must be given (with/without semicore electrons). The same applies for the value of the broadening.
- 3) Legend of Figure 5: "The energy axis is scaled" instead of "The energy axis of scaled". "Burgers vector" instead of "Burges vector". The colors of Figure 5b must match the code color chosen in the other figures (Δ dip in blue, unrelaxed energies in red), which will improve the clarity of the figure.
- 4) Details given for the calculations of the screw dislocation are contradictory: The authors write that the cell "has a repeat length of one Burgers vector along the dislocation line direction" and further in the text that "the supercell has been doubled along the z-axis". Also, the y-direction of the supercell used for dislocation calculations is different in the text ($[-110]$) and in Figure S1 ($[-1-10]$).
- 5) One can expect important surface contribution in the solute-dislocation interaction energy calculations within the cluster approach with flexible boundary conditions. How is this artefact taken into account?

Reviewer #2 (Remarks to the Author):

This paper addresses the origin of the interaction energy between substitutional solutes and defects in body centered cubic refractory metals, based on routine DFT calculations performed mainly for 1 matrix (W), 2 transition metal solutes (Re and Pt) and a variety of defects (vacancy, screw dislocation, grain boundary, twin boundary, generalized stacking faults(GSF)). GSF calculations are also performed for several solutes (Ta, W, Re, Os, Ir, Pt, and Au) in Ta and W. A parameter, Delta-dip, accounting for the change in bimodality of the local electronic density of states is proposed based on Hartigan's dip test statistic for unimodality. In the case of W a linear correlation is claimed between the unrelaxed interaction energy and Delta-dip. The origins of this correlation and the valence dependence of the slope are qualitatively explained on the basis of d-band energy.

The main finding of the present paper is summarized in Figs. (4) with a rather convincing average correlation between Delta-dip and unrelaxed binding energies in W for 2 solutes, Re and Pt. The Delta-dip parameter, proposed as an improved alternative to the fourth moment, is new and interesting. However I am not convinced by the robustness and generality of this result.

First the authors neglected to mention that this study is restricted to substitutional solutes, and more specifically to transition metal solutes.

This correlation is shown on unrelaxed binding energies, based on the argument that the relaxation energy is only a small correction as shown in Fig. 1 in the case of dislocation in W. But it is well known that lattice relaxation is particularly small in group VI metals (Mo and W) compared to group V (V, Nb, Ta). Relaxation effects are expected to become even dominant in group IV (in the high temperature bcc phase). I therefore don't expect this correlation to be relevant on relaxed binding energies in general.

This correlation is indeed very good for GSF, but some important deviations are observed, e.g. for the vacancy in W-Re. In the latter case, the binding energies are similar for first and second neighbors whereas Delta-dip varies by a factor of 7. It is surprising that self-interstitials, which have been investigated in several solute-defect DFT papers, have not been included in the present set of defects, which consists mostly of planar defects. I am not convinced that this correlation is as good for point defects as for planar defects.

This correlation is only demonstrated in the case of W matrix and for 2 solutes. This is by far not sufficient to claim universality. Similar calculations should be performed also at least in Ta (or another group V metal) for the same set of defects (and not just GSF). Moreover, the 2 solutes, Re and Pt, are both 5d solutes with more valence electrons than W. Universality should be tested also at least on solutes with less d electrons (see below), Hf and Ta.

A qualitative explanation for this correlation is proposed based on band energy arguments which are often used within tight-binding theory. It would be much more convincing to actually compute these band energy differences and see if they can quantitatively account for the observed interaction energies. Moreover, this qualitative explanation obviously fails for the case of the interaction of Hf and Ta solutes with dislocations in W which has been investigated by the same first author in Ref. 9. The present explanation predicts attractive interactions whereas the DFT calculations yield repulsive interactions.

Miscellaneous:

- The present interaction energies are calculated by taking as reference for infinitely separated defects the supercell with the two defects as far as possible from each other. This is probably a good approximation in most cases, but the standard in the community is to calculate it from a double difference on 4 cells (bulk, bulk+solute, defect, defect+solute).
- "these solute atoms prefer to occupy the core site instead of the bulk site due to their higher d-

band filling fraction": this sentence is misleading; I doubt that in a model where one can compute the energy per atom (eg tight-binding, not DFT) the energy would be lower in the core than in the bulk; on the other hand it can be shown with TB that differences in the fourth moment of the LDOS will induce two nodes in the band energy difference as function of band filling (i.e. the sign changes eg from positive for nearly empty band, to negative for half-filled, to positive for nearly filled, see papers by Ducastelle et al.).

- "the Δ dip parameter ... can serve as a useful indicator for the solute defect interaction energies ... without performing any calculations with substitutions of solute atoms": this sentence is a bit misleading; this is only true once the slope of the correlation has been established.

- I'm not sure "band-filling energy" is correct, I suggest "band energy"

- I'm not sure "half-full band" is correct; I suggest replacing by "half-filled band"

- "Hartigan's dip test was performed (See supplemental material for details) » -> "see method section for details"

- The value of the broadening width used for Methfessel-Paxton method must be specified

- Why is Fig. 2(b) shown for Os and not Re, which is studied afterwards ?

- The captions of Figs. 4 and 5 should explicitly mention that the interaction energies are unrelaxed

- The dislocation calculations are performed using a single dislocation and flexible boundary conditions; this method is well suited for calculating e.g. Peierls stresses, but computing solute-dislocation interaction energies assumes cancellation of errors of the energies at the boundaries

- "It should be noted that the supercell been doubled along the z-axis": this sentence seems to contradict "The supercell has a repeat length of one Burgers vector along the dislocation line direction". Have the calculations been performed over 1b or 2b?

Although the correlation between the present solute-defect interaction energies in W and Hartigan's dip test variation is new and interesting, the present results are far from being sufficiently robust and universal to deserve publication in Nature Communications.

Reviewer #3 (Remarks to the Author):

This is an interesting manuscript where the authors have investigated the correlation between solute-defect interactions and the deviation from bimodality in the band structure (local DOS) that is introduced in defect sites. The authors argue that the correlation between the interaction energies and a "dip parameter" is universal for bcc refractory metals and that this model can be used to rapidly screen for information during alloy design. It is a nice and well written manuscript and some of the conclusions are supported, but I find too many questions and unfortunately too little potential impact to recommend publication in Nature Communications.

I comment a few of the problematic issues with the manuscript:

Title: A "Universal correlation" is clearly not established here. Only a few cases have been studied and the correlations are (not that) clear and obvious. Only W and Ta among the bcc refractory metals have been studied.

* Abstract: "However, nowadays solute-defect interactions can only be accurately mapped case-by-case using extensive first-principles calculations, which limits our ability to manipulate alloy properties by tuning the interactions, especially for those related to defects at transition states." This is manifestly not correct. There are many other techniques than first-principles calculations. Experiments based on residual resistivity, muon spin rotation, positron trapping and annihilation, etc can be used to deduce solute-defect interactions. The authors should reformulate this sentence.

* Are the trends observed really going to speed up future calculations?

Method:

* The screw dislocation simulation cell is not suitable for detailed calculations of total energies. The fixed boundaries affect the different supercells (reference, different solute positions) so that the systematic error cancellation is no longer valid. This casts doubts over the results, unfortunately. For detailed investigation of dislocation interactions with solutes, impurities etc, the authors should refer to e.g. the many papers by Ventelon et al from CEA Saclay, France.

* $\Delta(\text{dip})$ is described in the end, but is not defined properly anywhere. This is hardly a canonical parameter that is well-known. The way it's calculated has to be clarified.

* "Moreover, the Δdip parameter of atoms near the defects in pure W configurations can serve as a useful indicator for the solute-defect interaction energies and the site occupation preference of the transition metal solute without performing any calculations with substitutions of solute atoms." -> This test should be performed then, showing how well it performs (or did I misunderstand something here)?

Results:

* Normally, the correlation error (figs 4) that give an indicative interaction energy from the $\Delta(\text{dip})$ is way too large - This level of precision can hardly be used for something like alloy property screening (even though the correlation itself is quite interesting and persuasive).

* In the correlation plots (Figs 4), the comparison of the $\Delta(\text{dip})$ and fixed interaction energies seem strongly biased to the 110-GSF results. This is a little worrisome, since there is no physical reason to have more such data points than anything else. Clearly the 112-GSF displays a different trend (very clear for Pt in W, still visible for Re in W). The authors don't comment this and it merits commenting. Also, what can quite often be of very high significance in alloy design, is reliable information on solute-vacancy interactions, and the correlation for that defect with the $\Delta(\text{dip})$ seems tenuous at best.

* Why do the authors only compare the static interaction energies? The relaxation effects can be quite important (even if the authors show how large this effect is for one case). Since the authors omit to display the correlation (or lack of such) for the relaxed interaction energies with the $\Delta(\text{dip})$, one wonders as to the reason. A strong correlation with relaxed interaction energies would be more persuasive (and useful!).

Reviewers' comments:

Reviewer #1 (Remarks to the Author):

The manuscript presents the results of DFT calculations of solute-defect interaction energies in body-centered cubic metals and proposes a correlation with the bimodality of the d -band local density of states close to the defect measured by the Hartigan's dip test through the Δdip parameter. The present work mainly focusses on rhenium and platinum substitutional solutes interacting with crystalline defects in bcc tungsten (vacancy, dislocation core, twin boundary, grain boundary, generalized stacking fault). A linear relation between the unrelaxed solute-defect interaction energy and the Δdip parameter is evidenced in tungsten. Based on these findings, supplemented by additional calculations of unrelaxed solute-GSF interaction energy in tungsten and tantalum for several 5d elements with a higher number of d electrons, a universal link between the Δdip parameter and the unrelaxed interaction energy is claimed by the authors. Although this correlation is very interesting in Figure 4, from my point of view it is not sufficiently proved to be general and strong to be accepted to be published in Nature Communications.

Answer: We are grateful for the reviewer's appreciation and valuable suggestions. To address the reviewer's primary concern ("*not sufficiently proved to be general and strong*"), we investigated the solute-defect interactions in a more broad range of matrix-solute element pairs (W-Pt, W-Re, W-Ta, Ta-Hf, and Ta-Os systems) and more types of defect structures (i.e. vacancy, self-interstitials at different sites, screw dislocation, twin boundary (TB) and grain boundaries (GB) with different orientations). According to our new results, the variation in the d -band bimodality still plays a significant role in characterizing the solute-defect interaction behaviors in bcc refractory metals, but the contribution from the valence s/p electrons also needs to be taken into account to yield a universally quantitative description on the interaction energy. A more accurate and robust linear correlation relationship solely based on two electronic factors is proposed by considering both the effects of the d -band bimodality and the hybridization between the sp - and d -bands (Fig. R3). We also provided a point-by-point response to your comments along with a detailed description of the changes in response to these comments below.

(1) This study is limited to substitutional solutes and this must be explicitly mentioned in the text.

Answer: Thanks for the comment. In the revised abstract and manuscript, it has been clearly mentioned that the solute-defect interactions studied in the present work were for substitutional cases. In addition, since our current model also considers the effect of $sp-d$ hybridization, similar studies on solute-defect interactions can be extended to interstitial solutes, such as C, N and O, where the interactions between s/p electrons in the solutes and d electrons in the metals may play a major role. We will perform these studies in the next step.

(2) Meanwhile, the correlation is established for various defects in the case of unrelaxed interaction energies in W and the fact that the energy is unrelaxed has to be more explicitly written.

Answer: Thanks for the comment. To avoid misleading and misunderstanding, the corresponding contents have been revised to mention that the fixed-lattice interaction energy

(i.e. unrelaxed interaction energy) is used for establishing the correlation with the electronic factors.

Meanwhile, we think it is necessary to explain the exact meaning of the “*unrelaxed interaction energies*” used in our manuscript. In the present work, the atomic configurations of the defect structures in the pure metals were initially fully relaxed. Then the “*unrelaxed interaction energies*” E_{int}^{fix} were calculated based on the fully relaxed defect structures in pure metals without further relaxations of atomic positions due to solute substitutions. In addition, the “*relaxed interaction energies*” (E_{int}^{relax}), obtained by fully relaxation of atomic positions due to solute substitutions, were also calculated for a few of the defect sites that have relatively strong ($|E_{int}^{relax}| > \sim 0.15$ eV) interactions with the solutes.

As shown in Fig. 1, Table S4 and S5 of the revised manuscript, the differences between E_{int}^{relax} and E_{int}^{fix} are quite small for these defect sites not only in W but also in Ta (details are discussed in the reply to the comment #4 of Reviewer #1). Thus, the relatively strong defect-solute interactions in bcc refractory metals may mainly originate from the changes in local electronic structures near the defects sites rather than the relaxation of the atomic positions upon the substitution of a solute atom. Since these strong interactions should play more important roles in mechanical and thermal properties of alloys than those weaker interactions, we directly calculated E_{int}^{fix} without considering the minor effects of the lattice relaxation. This choice of E_{int}^{fix} instead of E_{int}^{relax} can also speed up the calculation process and clearly reveal the quantitative correlation between the variation of interaction energy and the variation of local electronic structures due to solute substitutions.

(3) To support the universality of their correlation, the authors must perform a more systematic set of calculations in at least one group V element (and not only generalized stacking faults in Ta) and for other 5d solutes with a lower number of d electrons in addition to the solutes with a higher number of d electrons than in W presented in the paper. Based on the qualitative energy band analysis of Fig. 2 for the correlation with the variations in the bimodality of the LDOS, one would expect attractive binding energies in the case of i.e. Hf or Ta solutes in W.

Answer: We thank the reviewer for the helpful advice and suggestion. A short answer is that we have found a more accurate and robust linear correlation relationship solely based on two electronic factors to describe the general cases of solute-defect interactions in both Ta and W metals. More details are described as follows.

First, we performed many additional calculations to further testify the observed correlation between *d*-band bimodality and solute-defect interactions in bcc Ta and W. Specifically, the solute-defect interactions in matrix-solute element pairs of Ta-Hf, Ta-Os and W-Ta were additionally investigated. In addition to our previously studied crystalline defects, the DFT calculations were extended to cover more types of defect structures, including the $\langle 100 \rangle$ -dumbbell and $\langle 111 \rangle$ -dumbbell self-interstitials, and the $\Sigma 3$ (111) and $\Sigma 5$ (210) GBs, to testify the generality of the observed correlation.

Second, we confirmed that the *d*-band bimodality still plays a significant role in characterizing the solute-defect interaction behaviors in the cases of substitutional solute atoms with a lower number of *d* electrons than matrix elements. Here two related examples

are shown in Fig. R1a and R1b.

Fig. R1a plots the fixed-lattice interaction energies E_{int}^{fix} between Ta solutes and the $\Sigma 3(11\bar{2})$ TB in bcc W. Similar to Fig. 3 in the revised manuscript, E_{int}^{fix} and Δdip of each substitutional site are plotted with respect to the distance from the atomic site to the twin interface. It should be noted that the axis value for the Δdip term is plotted in reverse order (more positive values of Δdip toward the downward direction). As shown in Fig. R1a, the $\Sigma 3(11\bar{2})$ TB in W generally interacts with Ta solutes in a repulsive way ($E_{int}^{fix} < 0$ according to our definition in Fig. 1 in the manuscript). In contrast to Re and Pt, the interaction energy of Ta seems to show a weak and inverse correlation with the variation of the Δdip term. This can be explained by a shift in the position of the Fermi level on the local d -band of the solute atom relative to that of the W matrix atom, as illustrated in Fig. R2.

Figure R1. (a) Interaction of W $\Sigma 3(11\bar{2})$ TB with Ta solutes. The fixed-lattice interaction energy (E_{int}^{fix}) and the Δdip term of each atomic site are plotted with respect to the relative distance from the atomic site to the twin interface. (b) Interaction of W $\Sigma 5(310)$ GB with Ta solutes. E_{int}^{fix} and the Δdip term of each atomic site are defined in the same ways as those in (a).

Figure R2. (a) d -orbital LDOS of a Ta atom when it occupies the defect and reference site in the supercell of the W $\Sigma 3(11\bar{2})$ TB; (b) d -orbital LDOS of a W atom when it occupies the same atomic sites as the Ta atom in (a). The energy axis of both figures is scaled relative to the Fermi energy of each calculation.

As shown in Figs. R2a and 2b, the relative position of the Fermi level on the LDOS of Ta is moved towards to the lower energy state compared to that of W, no matter the Ta atom is placed at the defect or reference site. However, the shift in the position of Fermi level is quite

small since Ta has only one d electron less than W. As a result, in the W-Ta system, the Fermi level on the LDOS of the Ta solute in W matrix is located to a position which is even closer to the minimum of the pseudo-band gap compared to the W atom. According to our previous discussion in Fig. 2 of the manuscript, Ta atom should be even less preferred by the defect site than W atom by considering occupied states close to and far below the Fermi level. Therefore, the Ta solute atom interacts in a repulsive manner with the W $\Sigma 3(11\bar{2})$ TB generally ($\Delta dip > 0 \rightarrow E_{int}^{fix} < 0$). Similar results have also been found in other defect structures of bcc W. Fig. R1b shows another example: the interactions between Ta solute and the $\Sigma 5(310)$ GB in W.

Third, as the reviewer concerned, the correlation between E_{int}^{fix} and Δdip in W-Ta systems is found to be only qualitative and much weaker compared to the results of the W-Re and W-Pt systems. Especially, large discrepancies can be seen for several individual sites near the defects shown in Fig. R1. For example, E_{int}^{fix} at site 1 in Fig. R1b has a positive value (+0.147 eV, indicating an attractive behavior), but a repulsive interaction should be expected for the Ta-occupation if considering the corresponding Δdip value only (+0.022). Similarly, in the case of $\Sigma 3(11\bar{2})$ TB (Fig. R1a), the interaction energy of site 1 is about 5 times smaller than that of site 2, while the values of Δdip between these two sites only vary about 20%. In addition, sites 4 and 5 in $\Sigma 3(11\bar{2})$ TB shown in Fig. R1a have nearly zero values of Δdip and notable values of E_{int}^{fix} in contrast. All these discrepancies indicate that, in addition to the contribution from the filling energy of the d -band, there are other energy contributions to the solute-defect interactions in bcc W, which cannot be described by the Δdip term.

Fourth, to reconcile the above discrepancies, we propose a more accurate and robust linear correlation relationship solely based on two electronic factors by considering the effects of both the d -band bimodality and the hybridization between the sp - and d -band. The details of the results and the corresponding discussions can be found in the revised manuscript and supplementary materials. Here we just briefly summarize the key discussions and results as the following.

According to an energy band model developed by Hodges¹, Mueller² and Pettifor^{3,4}, the strength of the hybridization between the valence sp - and d -band in transition metal elements is proportional to the width of the local d -band. Furthermore, the local d -band width of an atom is closely related with the magnitude of the d - d interaction matrix elements (V_{dd}^{ij}) between the atom and its neighboring atoms^{5,6}. The relationship is written as^{5,6},

$$W_i \propto \sum_j V_{dd}^{ij} \quad \text{Eq. R1}$$

where j represents the neighboring atoms of atom i , W_i is the local d -band width and V_{dd}^{ij} is the d - d interaction matrix element between i and j , which can be scaled as^{6,7},

$$V_{dd}^{ij} \propto \frac{r_{d_i}^{\frac{3}{2}} r_{d_j}^{\frac{3}{2}}}{d_{ij}^5} \quad \text{Eq. R2}$$

where d_{ij} is the interatomic distance between atom i and j , and r_{d_i} is the spatial extent of d -orbital of atom i , which is an intrinsic element-property⁶.

The above derivation suggests the effects of the sp - d hybridization should vary with the atom at each individual defect site since d_{ij} of the atom can be different and the r_{d_i} of the solute element can differ from that of the neighboring matrix element. Thus, we consider the solute-defect interaction energy (E_{int}^{fix}) in two parts as shown in Eq. R3,

$$E_{int}^{fix} = \Delta E_d + \Delta E_{sp} \approx a_1 \Delta dip + a_2 x_{sp} \quad \text{Eq. R3}$$

Here, ΔE_d represents the energy contribution due to the d -band filling, which can be correlated with Δdip and a fitting coefficient, a_1 . ΔE_{sp} represents the energy contribution related to the sp - d hybridization. We propose ΔE_{sp} may also be estimated through a fitting coefficient, a_2 , and a variable, x_{sp} , that can describe the local environment of the defect site related to the sp - d hybridization. Like the Δdip term, it is expected that x_{sp} can also be obtained from the DFT calculations for the defects in pure metals. Then, the alloying effects can be reflected by the fitting coefficient a_1 and a_2 .

In the present work, x_{sp} of an atom at a site near the defect is proposed to be,

$$x_{sp} = 1 - \frac{(V_{vor}^{def})^{-\frac{5}{3}} / \epsilon_{sp}^{def}}{(V_{vor}^{ref})^{-\frac{5}{3}} / \epsilon_{sp}^{ref}} \quad \text{Eq. R4}$$

where $V_{vor}^{def} / V_{vor}^{ref}$ is the Voronoi volume of the atom at the defect and reference site (perfect lattice), respectively, and $\epsilon_{sp}^{def} / \epsilon_{sp}^{ref}$ is the center of the occupied sp -band projected on the atom at the defect and the reference site, respectively. In the present work, Voronoi volume of each atomic site is calculated from the relaxed atomic structures of pure matrix metals that contain defects. The ϵ_{sp}^{def} term is calculated as,

$$\epsilon_{sp}^{def} = \int_{-\infty}^0 E \rho_{sp}^{def}(E) dE / \int_{-\infty}^0 \rho_{sp}^{def}(E) dE \quad \text{Eq. R5}$$

where $\rho_{sp}^{def}(E)$ is the projected LDOS of the sp -band on the atom at the defect site and the Fermi energy is set as zero. ϵ_{sp}^{ref} is calculated in the same way for the atom at the reference site.

In Eq. R4, Voronoi volume (V_{vor}) is used to describe the average changes in the interatomic distances of the atoms near the defect, and the resulting changes in the d - d interaction matrix elements (V_{dd}^{ij}) and effects on the local d -band width (W_i). The exponent value, $-\frac{5}{3}$, is obtained based on Eq. R2 that V_{dd}^{ij} is proportional to $(d_{ij})^{-5}$. A benefit of using Voronoi volume instead of directly calculating d_{ij} is to avoid arbitrary assignment of the cutoff distance for identifying neighboring atoms. In Eq. R4, we also include a scaling term that is calculated as $1/\epsilon_{sp}$. Based on the definition (Eq. R5), ϵ_{sp} actually characterizes the average difference between the energy of the sp -band and the Fermi level. Generally speaking, the electrons with energies close to the Fermi level may be more sensitive to small perturbations. We thus propose that for the sp -band closer to the Fermi level its corresponding sp - d hybridization may have a stronger effect on the ground state energy. Therefore, the inverse of ϵ_{sp} is used to scale the effects of sp - d hybridization on solute-defect interactions. It is noteworthy that the inclusion of the ϵ_{sp} term in Eq. R4 is necessary as we found that the

Voronoi volume alone is inadequate to construct the x_{sp} term to yield an accurate description of the solute-defect interaction energy. In addition, attempts have been made to adopt the electronic factor descriptors based on other common features of the electronic bands, such as the d -band center, zero-, first- and second-order moments of the sp - and d -band and so on. However, Δdip and x_{sp} are the two descriptors so far that yield the most accurate regression results.

Based on Eq. R3, linear regressions are performed to model the DFT-calculated E_{int}^{fix} of the defects in the W-Ta, W-Re, W-Pt, Ta-Hf and Ta-Os systems. Δdip and x_{sp} are treated as regression variables, which are obtained for each defect site from the corresponding DFT defect-calculations in pure metals. a_1 and a_2 are fitting coefficients, which should be varied for each alloy system to reflect the effects of the solute elements. The least squares fitting method is employed to perform the regression. The regression parameters for each matrix-solute element pair are summarized in Table R1. Figs. R3 shows the parity plots of the solute-defect interaction energies (E_{int}^{fix}) from the direct DFT calculations and the predictions of the regression model.

Figure R3. Comparison between the E_{int}^{fix} from direct DFT calculations and predicted from the regression model in based on Eq. R3. (a) W-Ta system: Ta as solute in W matrix; (b) W-Re system: Re as solute in W matrix; (c) W-Pt system: Pt as solute in W matrix; (d) Ta-Hf system: Hf as solute in Ta matrix; (e) Ta-Os system: Os as solute in Ta matrix. The data of the point, line and planar defects are marked in circle, triangle and square symbols, respectively.

Table R1. Regression coefficients and parameters of the linear regression model based on Eq. R3.

Alloy system	coefficient		Adjusted R^{2*}	Standard error	p-value**	
	a_1	a_2			Δdip	x_{sp}
W-Ta	-7.18	1.78	0.9252	0.044	1.5e-23	1.3e-31
W-Re	15.98	-1.29	0.9603	0.039	1.6e-56	2.0e-30
W-Pt	61.80	-1.00	0.9453	0.168	3.4e-53	1.0e-03
Ta-Hf	6.66	2.08	0.9347	0.043	2.5e-19	6.0e-35
Ta-Os	-6.43	-4.06	0.8848	0.107	4.6e-06	5.3e-29

*The adjusted R^2 is the coefficient of determination, which represents the proportion of the variance in the regression response that is predictable from the regression variables. **The p-value tests the null hypothesis for each regression variable. A small p-value (typically ≤ 0.05) indicates strong evidence to reject the null hypothesis, which means the changes in the variable value (Δdip and x_{sp}) is strongly related to the changes in the fitting response value (E_{int}^{fix}).

As shown in Fig. R3, the solute-defect interaction energies (E_{int}^{fix}) predicted by the proposed model show good agreement with the results of DFT calculations for both Ta and W alloys with different substitutional transition metal solutes. Good regression quality is also demonstrated by the close-to-one value of adjusted R^2 and a small value of standard error, which is about 5%~10% of the largest response value for each regression as listed in Table R1. In addition, it is found that the p-values of both variables (i.e. Δdip and x_{sp}) are almost zero, significantly less than 0.05, which means both variables are statistically meaningful to explain the response variance. Comparing the present regression results of the W-Re and W-Pt systems to the results from our old model (Fig. S7 in the revised manuscript), one can find that both of the present and old regression models yield a very close value for the fitting

coefficient of Δdip but the present model has a higher adjusted R^2 and a smaller standard error. This indicates that the residual variance in E_{int}^{fix} that cannot be explained by the Δdip term is approximately explained by the x_{sp} term.

Moreover, as reflected by the sign of the coefficient a_1 , in the W alloys, the Δdip term yields a positive contribution to interaction energy for the solute with more d electrons than W (e.g. Re and Pt), and gives a negative contribution for the solute with less d electrons (e.g. Ta). In contrary, the contribution becomes positive in the Ta alloys for the solute with less d electron than Ta (e.g. Hf) while negative for the solute with slightly more d electron (e.g. Os). This is consistent with the analyses of Fig. 5 in the previous manuscript (also discussed under the subsection “Disparities and similarities between group V and VI bcc matrix elements” in revised manuscript), that is, the relative position of the Fermi level on the LDOS of the d -band is intrinsically different between Ta and W. After alloying, the relative position of Fermi level on the LDOS of solute atoms can either be shifted away or moved towards to the minimum of the pseudo-band gap due to solute-induced changes in d -band filling fraction. Consequently, a reduction in the d -band bimodality can either yield a positive or negative contribution to the interaction energy.

Additionally, it is found that the sign of the coefficient a_2 also correlates with the number of d electrons of the solute element. As shown in Table R1, a_2 generally has a positive sign if the solute element has less d electrons than the matrix element (e.g. W-Ta and Ta-Hf), while a negative sign if the difference in the number of d electrons is reversed. This correlation can be understood in terms of the difference in the spatial extent of d -orbital between solute and matrix elements. Details are explained in Section 9 in our revised supplementary materials.

Finally, since the manuscript was extensively revised and reorganized to incorporate these new findings, we have changed the manuscript title to “**Universal correlation between electronic factors and solute-defect interactions in bcc refractory metals**” to accurately reflect the conclusions of this study.

(4) Based on their calculations in tungsten, the authors assume that the effect of relaxation on the solute-defect interaction energies is small enough to be neglected further in the work. However, the authors might know that the group dependence of the relaxation around a defect can be important, and in particular it is well-established that relaxations are more important for group V elements than for group VI elements.

Answer: Thanks for the comment and we agree with the reviewer. Generally speaking, relaxation is more important for the defect energetics for group V elements than for group VI elements. For example, as indicated in the work by Ventelon et al.⁸, relaxations on atomic positions relative to their positions in perfect lattice are crucial for calculating the formation energy of the mono-vacancy defect in bcc Ta. As explained in our reply to comment #2 of Reviewer #1, in the present work, the initial ideal configurations of the defect structures were indeed fully relaxed before solute substitution to obtain the optimized defect structure in pure matrix metals. Then, the solute substitution is based on the optimized defect structures instead of the initial ideal atomistic structures without any relaxations.

Our previous calculations (Fig. 1 and Table S5 in the revised manuscript) showed that the

difference between the relaxed (E_{int}^{relax}) and fixed-lattice (i.e. unrelaxed) defect-solute interaction energies (E_{int}^{fix}) is small for W alloys. Here we show that this may be true for Ta alloys as well. In the revision, E_{int}^{relax} and E_{int}^{fix} in the Ta-Hf system are calculated for a few of defect sites that have relatively strong interactions with the solutes. As shown in Table R2, the difference between E_{int}^{relax} and E_{int}^{fix} also seems to be small in Ta alloys. This indicates that no matter if it is Ta- or W-based transition metal alloys, the defect-solute interactions mainly originate from the local changes in electronic bonding near the defect sites rather than the energy change by the relaxation of the atomic positions.

In addition, an expected application of the correlations between E_{int}^{fix} and the electronic factors found in our manuscripts will be to speed up the search and design of alloy defects in the large parameter space of the compositional and structural features. For such purposes, the less accurate E_{int}^{fix} can be utilized for the initial search/design steps, and the more accurate E_{int}^{relax} can be obtained in a parameter space that has been significantly narrowed down. Thus, any methods for fast prediction of E_{int}^{fix} will be useful to future alloy design.

Table R2. Calculated E_{int}^{relax} and E_{int}^{fix} of the interactions between Hf and the crystalline defects in bcc Ta. (unit: eV)

	Vacancy	<111>- dumbbell	$\Sigma 3$ ($11\bar{2}$) twin boundary	$\Sigma 5$ (310) grain boundary
Site*	1-nn	2-nn	1-nn	1-nn
E_{int}^{relax}	0.388	0.213	0.138	0.499
E_{int}^{fix}	0.351	0.223	0.152	0.489

*1-nn: the atomic site with first shortest distance to defect center; 2-nn: the atomic site with second shortest distance to defect center

Corresponding revision has been made in the main text of the manuscript. Table R2 is also added as Table S4 in the revised supplementary materials.

Some remarks:

(5) The solute-GSF interaction energy has been calculated based on the commonly used definition: $E(\text{defect+solute})+E(\text{bulk})-E(\text{defect})-E(\text{solute})$. Why is this definition not used by the authors for the other defects instead of calculating the energy difference between the solute close to the defect and as far as possible from the defect?

Answer: Thanks for the comment. The four-supercell method is usually considered to yield a reference state where solute atom is infinitely separated from the defect. Since the present work focuses on the solute-defect interaction induced by the local electronic features of the atoms near defect, we would like to connect the solute-defect interaction energies with the difference in the electronic structures between a near-defect atom and an atom with a finite distance to the defect center. We expect this would be a more persuasive way to show that the interaction behavior is mainly caused by variations in the local electronic structures instead of

a long-range elastic solution, which can be obtained easily based on classical elastic methods. In addition, as shown in Fig. 3 in the revised manuscript, the calculated interaction energy is also well converged with respect to the separation distance. Furthermore, we used the four-supercell method for the GSF calculations is due to a previous concern on the relatively small size of the supercell. In the revision, we have tested the two-supercell method for the GSF calculations, which gives very similar values for the interaction energies. Therefore, in the revision, it is unified to show the results from the calculations using the two-supercell method.

(6) Details about the pseudopotentials used in the present study must be given (with/without semicore electrons). The same applies for the value of the broadening.

Answer: Thanks for the suggestion. The electronic configurations of the pseudopotentials used in the present first-principles calculations are included as Table S1 in the revised supplementary. In addition, the value of the broadening width is also specified in the methodology section.

(7) Legend of Figure 5: “The energy axis is scaled” instead of “The energy axis of scaled”. “Burgers vector” instead of “Burges vector”. The colors of Figure 5b must match the code color chosen in the other figures (Δdip in blue, unrelaxed energies in red), which will improve the clarity of the figure.

Answer: Many thanks for pointing these typos out. Fig. 5 in the previous manuscript has been modified and moved to supplementary materials. In the revised manuscript, all the figure legends are proofread to avoid similar typos.

(8) Details given for the calculations of the screw dislocation are contradictory: The authors write that the cell “has a repeat length of one Burgers vector along the dislocation line direction” and further in the text that “the supercell has been doubled along the z-axis”. Also, the y-direction of the supercell used for dislocation calculations is different in the text ([-110]) and in Figure S1 ([$1-10$]).

Answer: We apologize for the misleading statement. In fact, the relaxation calculation for the dislocation structure in pure W is performed using the supercell with the z-axis length as one Burgers vector. Then, the relaxed structure is doubled along the z-axis to make a new supercell that is used for the calculations of solute-defect interactions. The purpose of doing so is to minimize the possible solute-solute image interactions due to the periodic boundary condition in the DFT calculations.

The corresponding statements in the supplementary are revised to elaborate the calculation scheme more clearly. Fig. S1 (Fig. S2 in the revised supplementary) is also modified to match the text.

(9) One can expect important surface contribution in the solute-dislocation interaction energy

calculations within the cluster approach with flexible boundary conditions. How is this artefact taken into account?

Answer: Thanks for the comment. The supercell in the present work was built using the domain boundary method⁹, which does not contain a free surface. The effects of the domain boundary on the energy calculations are discussed in the reply to the comment #4 of Reviewer #3.

Reviewer #2 (Remarks to the Author):

This paper addresses the origin of the interaction energy between substitutional solutes and defects in body centered cubic refractory metals, based on routine DFT calculations performed mainly for 1 matrix (W), 2 transition metal solutes (Re and Pt) and a variety of defects (vacancy, screw dislocation, grain boundary, twin boundary, generalized stacking faults(GSF)). GSF calculations are also performed for several solutes (Ta, W, Re, Os, Ir, Pt, and Au) in Ta and W. A parameter, Delta-dip, accounting for the change in bimodality of the local electronic density of states is proposed based on Hartigan's dip test statistic for unimodality. In the case of W a linear correlation is claimed between the unrelaxed interaction energy and Delta-dip. The origins of this correlation and the valence dependence of the slope are qualitatively explained on the basis of d-band energy. The main finding of the present paper is summarized in Figs. (4) with a rather convincing average correlation between Delta-dip and unrelaxed binding energies in W for 2 solutes, Re and Pt. The Delta-dip parameter, proposed as an improved alternative to the fourth moment, is new and interesting. However I am not convinced by the robustness and generality of this result.

Answer: We appreciate the reviewer's careful review and valuable suggestions. To further evaluate the robustness and generality of the observed correlation, we investigated the solute-defect interactions in a more broad range of matrix-solute element pairs (W-Pt, W-Re, W-Ta, Ta-Hf, and Ta-Os systems) and more types of defect structures (i.e. vacancy, self-interstitials at different sites, screw dislocation, twin boundary (TB) and grain boundaries (GB) with different orientations). According to our new results, the variation in the *d*-band bimodality still plays a significant role in characterizing the solute-defect interaction behaviors in bcc refractory metals, but the contribution from the valence *s/p* electrons also needs to be taken into account. The effects of the valence *d* and *s/p* electrons lead to two key electronic factors, namely (i) the *d*-band bimodality and (ii) the strength of *sp-d* hybridization. A more accurate and robust linear correlation relationship solely based on the two electronic factors is proposed and demonstrated to be able to quantitatively describe the solute-defect interaction energies in all the investigated Ta- and W-based alloys. (Results are summarized in Fig. R3 in this reply letter). In the following, we also provide a point-by-point response to your comments along with a detailed description of the changes in response to these comments.

- (1) First the authors neglected to mention that this study is restricted to substitutional solutes, and more specifically to transition metal solutes.

Answer: Thanks for the comment. In the revised manuscript, it has been clearly mentioned that the solute elements studied in the present work are transition metals and they are treated as substitutional atoms. In addition, based on our new calculation results, we propose a new model to characterize the substitutional solute-defect interactions in bcc refractory metals based on not only the variation in the d -band bimodality but also the contribution from the hybridization between the valance sp - and d -bands. Such model can be possibly extended for the future studies on interstitial solutes, such as C, N and O, where the interactions between s/p electrons in the solutes and d electrons in the metals may play a major role. We will perform these studies in the next step.

- (2) This correlation is shown on unrelaxed binding energies, based on the argument that the relaxation energy is only a small correction as shown in Fig. 1 in the case of dislocation in W. But it is well known that lattice relaxation is particularly small in group VI metals (Mo and W) compared to group V (V, Nb, Ta). Relaxation effects are expected to become even dominant in group IV (in the high temperature bcc phase). I therefore don't expect this correlation to be relevant on relaxed binding energies in general.

Answer: Thanks for the comment. This comment is related to the comment #4 of Reviewer #1 and please check our detailed reply there. We also addressed this issue in the revised supplementary materials (*section 4*).

- (3) This correlation is indeed very good for GSF, but some important deviations are observed, e.g. for the vacancy in W-Re. In the latter case, the binding energies are similar for first and second neighbors whereas Δdip varies by a factor of 7.

Answer: We really appreciate the reviewer for pointing out this issue, which inspired us to realize the presence of other unrevealed mechanisms contributing to the solute-defect interactions in bcc refractory alloys. In addition, we also took this opportunity to double-check our calculations. We apologize that there is indeed a mistake of inconsistency in the calculations of vacancy-solute interactions. Specifically, the calculation of defect electronic structure was performed for a vacancy configuration obtained from a calculation in which only atomic relaxations are performed and the supercell cell lattice is unrelaxed and fixed as the equilibrium bulk lattice. However, the solute-vacancy interaction energies are calculated in a fully relaxed defect configuration (both atomic position and supercell geometry are fully relaxed) in pure W metal (without further relaxation due to solute substitution). In the revision, we re-calculated the electronic structure of the vacancy defect using a fully relaxed configuration in pure W metal and updated Fig. 3a in the revised manuscript accordingly.

As shown in the updated Fig. 3a, Δdip of the first-nearest-neighbor site now has a value about three times larger than that of the second-nearest-neighbor site, which is still very different from the variation in the interaction energies (E_{int}^{fix}) between these two sites. This

large discrepancy indicates that there are other mechanisms contributing to the solute-defect interactions, which cannot be described by the Δdip parameter. As discussed in the reply to the comment #3 from Reviewer #1, the contributions from the valence sp -band have to be taken into account in order to adequately capture the interaction energies. As shown in Fig. R3b, after considering the contribution from the sp -band, the newly proposed model yields predictions of 0.201 and 0.164 eV for the interaction energies of the first- and second-nearest Re-vacancy pairs, respectively, which now agrees well with the results of DFT calculations (0.226 and 0.181 eV).

We have also double-checked the rest calculations of other crystalline defects to make sure that all the results are correctly reported.

- (4) It is surprising that self-interstitials, which have been investigated in several solute-defect DFT papers, have not been included in the present set of defects, which consists mostly of planar defects. I am not convinced that this correlation is as good for point defects as for planar defects.

Answer: Thanks for the reviewer's suggestion. In the revision, additional DFT calculations were performed to investigate the interactions between the solute elements and self-interstitial defects, namely, the $\langle 100 \rangle$ - and $\langle 111 \rangle$ -dumbbells. As shown in Fig. R3, E_{int}^{fix} of these point defects can be described by our newly proposed model along with the line and planar defects.

- (5) This correlation is only demonstrated in the case of W matrix and for 2 solutes. This is by far not sufficient to claim universality. Similar calculations should be performed also at least in Ta (or another group V metal) for the same set of defects (and not just GSF). Moreover, the 2 solutes, Re and Pt, are both 5d solutes with more valence electrons than W. Universality should be tested also at least on solutes with less d electrons (see below), Hf and Ta.

Answer: Many thanks for the comment. This comment is addressed together with the comment #3 from Reviewer #1.

- (6) A qualitative explanation for this correlation is proposed based on band energy arguments which are often used within tight-binding theory. It would be much more convincing to actually compute these band energy differences and see if they can quantitatively account for the observed interaction energies.

Answer: Many thanks for the suggestion. We agree that a more convincing illustration would be to directly compare the difference in band energy per atom if the band energy can be precisely calculated. However, in the VASP and most of other DFT codes, the LDOS on each atom is calculated by projecting the orbitals onto spherical harmonics that are non-zero within a spherical space around each atom. As a result, the LDOS obtained from DFT calculations is not a conserved quantity, which cannot be directly used for calculating the band energy. On the other hand, we assume a relative change in the shape characters of the

LDOS (e.g. bimodality) can still be well captured by this projection method as the spherical harmonics are the same for the same type of element in all calculations.

- (7) Moreover, this qualitative explanation obviously fails for the case of the interaction of Hf and Ta solutes with dislocations in W which has been investigated by the same first author in Ref. 9. The present explanation predicts attractive interactions whereas the DFT calculations yield repulsive interactions.

Answer: Many thanks for the comment. We apologize for the misleading. In fact, the repulsive interaction behaviors of Hf and Ta solutes with the screw dislocation in W could still be understood in terms of the local variations in the d -band bimodality and filling fraction near the defect.

When a W atom is substituted by a solute atom with fewer d electrons, the relative position of Fermi level on the LDOS of the solute atom should shift towards to the lower energy of the d -band compared to that of W. However, because of the charge transfer between the solute atom and the solvent W atoms, the solute atom Ta or Hf in the W-based alloy should have a higher filling fraction in the d -band compared to its counterpart in a matrix purely composed of the solute element itself. For example, as shown in Fig. 5c in the previous manuscript (also Fig. 5a in the revised manuscript), the Fermi level of Ta in pure Ta matrix is located far below the energy corresponding to the minimal position of the bcc pseudo-band gap. However, due to the charge transfer, the Fermi level on the LDOS of a Ta solute atom in W matrix is at an energy state different from that of pure Ta, as shown in Fig. R2a and Fig. 2d in the revised manuscript.

Furthermore, as shown in Fig. 2a in the revised manuscript, the Fermi level of pure W is located at an energy close to the minimum of the bcc pseudo-band gap, but not exactly at the minimal position. Therefore, after introducing the solute atom with fewer d electrons (e.g. Ta and Hf) in the defect site, the relative positive of Fermi level on the local d -band of the solute atom will locate at an energy even more closer to the minimum of the pseudo-band gap compare to that of W atom, as shown in Fig. 2d in the revised manuscript. As a result, the solute atom should be even less preferred by the defect site with less d -band bimodality (more positive Δdip) than W atom, which means a repulsive interaction between the solute and defect can be expected in terms of the effects from d -band bimodality and filling fraction.

We have revised the manuscript to include the above discussion for the solute elements with fewer d electrons. Together with the modifications of our previous corresponding statements, the revised manuscript now explains this mechanism more clearly.

Miscellaneous:

- (8) The present interaction energies are calculated by taking as reference for infinitely separated defects the supercell with the two defects as far as possible from each other. This is probably a good approximation in most cases, but the standard in the community is to calculate it from a double difference on 4 cells (bulk, bulk+solute, defect, defect+solute).

Answer: Many thanks for the comment. This comment is addressed together with the comment #5 of Reviewer #1.

- (9) “these solute atoms prefer to occupy the core site instead of the bulk site due to their higher d -band filling fraction”: this sentence is misleading; I doubt that in a model where one can compute the energy per atom (e.g. tight-binding, not DFT) the energy would be lower in the core than in the bulk; on the other hand it can be shown with TB that differences in the fourth moment of the LDOS will induce two nodes in the band energy difference as function of band filling (i.e. the sign changes eg from positive for nearly empty band, to negative for half-filled, to positive for nearly filled, see papers by Ducastelle et al.).

Answer: Many thanks for the comment. We agree with the reviewer that this sentence is misleading, which implies that the solute atom with higher d -band filling fraction may have a lower energy per atom at the core site than in the bulk. In fact, this may not be true and is not a necessary requirement for resulting in an attractive solute-defect interaction. If we only consider the contribution from the d -band energy, the interaction energy, E_{int}^{fix} , can be approximated as,

$$E_{int}^{fix} \approx (E_{sol}^{ref} + E_W^{def}) - (E_{sol}^{def} + E_W^{ref}) \quad \text{Eq. R6}$$

where E_{sol}^{def} and E_{sol}^{ref} represent the d -band energy per solute atom when it occupies the core and reference site, respectively, and E_W^{def} and E_W^{ref} are the d -band energy per W atom when it is in the core and reference site, respectively. Based on Eq. R6, the solute can have an attractive interaction with the dislocation (i. e., $E_{int}^{fix} > 0$) as long as the value of $E_{sol}^{ref} - E_{sol}^{def}$ is less negative than the value of $E_W^{ref} - E_W^{def}$, even though the solute itself may have a lower energy in bulk than in the core (i.e. $E_{sol}^{ref} < E_{sol}^{def}$). In addition, we also agree with the reviewer that a smaller fourth moment can also lead to a more destabilizing contribution for a nearly empty band. Thus, “higher d -band filling fraction” is also not a necessary condition to result in an attractive solute-defect interaction. Therefore, a more precise expression should be “*compared to placing W atoms at the core site, the system may benefit from a stabilization contribution from the band energy when the core site is occupied by the solute atom that results in the local filling fractions towards to the band edge*”. The corresponding changes have also been made in the revised manuscript.

- (10) “the Δdip parameter ... can serve as a useful indicator for the solute defect interaction energies ... without performing any calculations with substitutions of solute atoms”: this sentence is a bit misleading; this is only true once the slope of the correlation has been established.

Answer: Many thanks for the comment. We agree with the reviewer that this sentence is misleading. This comment is addressed together with the comment #6 of Reviewer #3.

(11)

- I'm not sure "band-filling energy" is correct, I suggest "band energy"

- I'm not sure "half-full band" is correct; I suggest replacing by "half-filled band"

- "Hartigan's dip test was performed (See supplemental material for details) » -> "see method section for details"

Answer: Many thanks for the suggestions. The corresponding words are revised in the manuscript.

(12) The value of the broadening width used for Methfessel-Paxton method must be specified

Answer: The value of the broadening width for DFT calculations is specified in the methodology section.

(13) Why is Fig. 2(b) shown for Os and not Re, which is studied afterwards ?

Answer: Many thanks for the comment. Both solute elements were studied in terms of their interactions with the $\frac{1}{2}\langle 111 \rangle$ screw dislocation in W. One of the purposes to show the LDOS of W, Os and Pt is to show how a gradual change in the number of *d* electrons affects the relative positions of the Fermi level on LDOS, since Os has two more *d* electrons than W and Pt has two more *d* electrons than Os.

(14) The captions of Figs. 4 and 5 should explicitly mention that the interaction energies are unrelaxed

Answer: Many thanks for the comment. We have changed the captions of the corresponding figures in the revised manuscript

(15) The dislocation calculations are performed using a single dislocation and flexible boundary conditions; this method is well suited for calculating e.g. Peierls stresses, but computing solute-dislocation interaction energies assumes cancellation of errors of the energies at the boundaries.

Answer: This comment falls into the same category as the comment #4 of Reviewer #3, which will be addressed with that comment together.

(16) "It should be noted that the supercell been doubled along the z-axis": this sentence seems to contradict "The supercell has a repeat length of one Burgers vector along the dislocation line direction". Have the calculations been performed over 1b or 2b?

Answer: This is related to the comment #8 of Reviewer #1. Please see our reply there. We

also revised the manuscript accordingly.

- (17) Although the correlation between the present solute-defect interaction energies in W and Hartigan's dip test variation is new and interesting, the present results are far from being sufficiently robust and universal to deserve publication in Nature Communications.

Answer: We appreciate the reviewer's acknowledge on the novelty and importance of our work. To address the reviewer's concerns on the "**robust and universal**" quality, we further investigated the solute-defect interactions behaviors in the W-Ta, Ta-Hf and Ta-Os systems in additions to W-Re and W-Pt systems. Besides, the DFT calculations were extended to cover more types of defect structures to validate the generality of the proposed correlation. Based on the calculation results, we found more "**robust and universal**" correlations between defect-solute interactions and the electronic features of defects in pure metals (Fig. R3 in the reply). For this reason, we hope our revised manuscript may deserve publication in Nature Communications.

Reviewer #3 (Remarks to the Author):

This is an interesting manuscript where the authors have investigated the correlation between solute-defect interactions and the deviation from bimodality in the band structure (local DOS) that is introduced in defect sites. The authors argue that the correlation between the interaction energies and a "dip parameter" is universal for bcc refractory metals and that this model can be used to rapidly screen for information during alloy design. It is a nice and well written manuscript and some of the conclusions are supported, but I find too many questions and unfortunately too little potential impact to recommend publication in Nature Communications.

Answer: We are grateful for the reviewer's appreciation and valuable suggestions. In the revised manuscript, an improved correlation relationship is introduced to yield a universal and accurate description on the solute-defect interaction energies for more generalized solute-matrix pairs (Results are summarized in Fig. R3 in this reply letter and also answered in detail in the reply to comment #1 of the reviewer). We also add more contents in the discussion section to further illustrate the potential impacts and future applications of the present work. In short, our study provides general and quantitative guidance on solute-defect interactions in alloys based on electronic structure descriptors, which can speed up the understanding and manipulation of stability and mobility of more complex but important defects (such as curved dislocations/grain boundaries with kinks/steps) in alloys with the help of machine learning techniques (details in the reply to comment #3 of the reviewer). Here we provide a point-by-point response to your comments along with the corresponding changes in response to these comments.

I comment a few of the problematic issues with the manuscript:

- (1) Title: A "Universal correlation" is clearly not established here. Only a few cases have been studied and the correlations are (not that) clear and obvious. Only W and Ta among the bcc refractory metals have been studied.

Answer: Many thanks for the comment. The bcc refractor metal elements generally refer to the group V and VI elements. Several previous works have shown that the elements in a same group usually have very similar solute-defect interaction behaviors when they form binary alloys with transition metal solute elements (e.g. Mo and W^{10,11}, Nb and V^{12,13}). This is because these interactions are mainly determined by the valance electrons. Therefore, in the present work, we choose Ta and W as two samples to represent other bcc refractory elements in group V and VI, respectively.

Moreover, we totally agree with reviewer that more cases of solute-defect interactions should be calculated to further evaluate the generality of the proposed correlation. To further evaluate the robustness and generality of the observed correlation, we investigated the solute-defect interactions in a broader range of matrix-solute element pairs (W-Pt, W-Re, W-Ta, Ta-Hf, and Ta-Os systems) and more types of defect structures (i.e. vacancy, self-interstitials at different sites, screw dislocation, twin boundary (TB) and grain boundaries (GB) with different orientations). According to our new results, the variation in the *d*-band bimodality still plays a significant role in characterizing the solute-defect interaction behaviors in bcc refractory metals, but the contribution from the valance *s/p* electrons also has to be taken into account. A more accurate and robust linear correlation relationship solely based on two parameters is proposed by considering both the effects of the *d*-band bimodality and *sp-d* hybridization. A detailed discussion is provided in the reply to the comment #3 of Reviewer #1. Corresponding revisions have also been made in the manuscript.

- (2) Abstract: "However, nowadays solute-defect interactions can only be accurately mapped case-by-case using extensive first-principles calculations, which limits our ability to manipulate alloy properties by tuning the interactions, especially for those related to defects at transition states." This is manifestly not correct. There are many other techniques than first-principles calculations. Experiments based on residual resistivity, muon spin rotation, positron trapping and annihilation, etc can be used to deduce solute-defect interactions. The authors should reformulate this sentence.

Answer: Thanks for pointing this out. The sentence has been removed in the revised manuscript.

- (3) Are the trends observed really going to speed up future calculations?

Answer: We appreciate that the reviewer for raising this question. The trends observed in this study can indeed speed up the calculation and research of solute-defect interactions and their effects. The discovered universal correlation not only reveals the underlying physics (electronic origin) of the solute-defect interactions in bcc refractory metals, but also provides a new way to efficiently map the interaction energy of each atomic site near the defect. For example, as shown in Fig. R4, with the correlation relationship established based on the calculations for a few of simple point and planar defects, one can estimate the interaction

energy for a complex defect structure (e.g. dislocation) with reasonably small uncertainty. In another word, instead of running many individual DFT calculations for different substitutional configurations, only one DFT calculation for the defect in pure metal configuration is required for obtaining the Δdip and x_{sp} parameters for each atomic site. This could significantly reduce the computational costs for defects in highly complex geometry.

More importantly, since variations in both the bimodality of d -band LDOS and the sp - d hybridization are governed by the local atomic structure of the defects, in the future, it is possible to quantitatively connect the Δdip and x_{sp} parameters of a defect site to its local structural features, such as the bond orientation parameter¹⁴, coordination number, radial distribution function and so on. Such connection can be achieved by applying the machine learning method^{15,16}. As a result, one would be able to estimate the electronic factors (i.e. Δdip and x_{sp}) of each atomic site in a defect geometry optimized from the atomistic simulations based on empirical interatomic potentials rather than first-principles calculations. Then, with the established correlation between the solute-defect interaction and the electronic factors, the variations in the energetic properties of the various types of crystalline defects due to transition-metal solutes can be efficiently predicted. This will be particularly useful for complex defect configurations critical to their kinetics, such as dislocation kinks/jogs and disconnections in grain boundaries^{17,18}, which are otherwise hard to directly study using first-principles calculations. In summary, we believe the observed correlation can be both scientifically and technically important for the future alloy design with targeted properties by controlling solute-defect interactions. These discussions have also been added in the revised manuscript.

Method:

- (4) The screw dislocation simulation cell is not suitable for detailed calculations of total energies. The fixed boundaries affect the different supercells (reference, different solute positions) so that the systematic error cancellation is no longer valid. This casts doubts over the results, unfortunately. For detailed investigation of dislocation interactions with solutes, impurities etc, the authors should refer to e.g. the many papers by Ventelon et al from CEA Saclay, France.

Answer: Many thanks for this comment. We agree with the reviewer that the dislocation-dipole method is widely used for the calculations of solute-dislocation interactions. However, it is also well-established that the flexible boundary condition method can give reliable predictions on solute-dislocation interaction energies for many alloy systems, such as Mg-based alloys¹⁹, Al-based alloys²⁰, Fe-based alloys²¹, Mo-based alloys¹⁰ and W-based alloys¹¹. In addition, the interactions between the transition metal solute and $\frac{1}{2}\langle 111 \rangle$ screw dislocation in W have also been studied using the dislocation-dipole method in a recent work by Tsuru et al.²² As shown in Table R3, their calculation results are in very good agreement with the interaction energies that calculated using the flexible boundary condition method¹¹.

Table R3. Comparison between the relaxed solute-dislocation interaction energies that calculated using the dislocation-dipole method and flexible boundary condition method in bcc W. (unit: eV)

	Re	Os	Ir	Pt
Dislocation-dipole method	0.23	0.62	1.14	1.28
Flexible boundary condition method	0.24	0.63	1.11	1.26

In fact, the supercell boundary has very limited effect on the calculations of the interaction energy due to the large size of the supercell. For example, in present work, the supercell has a length of the x and y vectors about 40 Å, while the distance between the core and reference site is only 10.6 Å. Since the dislocation geometry is placed in the center of the supercell, changing the solute position from core to reference site should have almost no effect on the boundary atoms. For example, we found that the changes in the Hellmann-Feynman force of the boundary atoms is only about 1~2 meV/Å on average. A detailed analysis of the boundary effects on the total energy calculation can also be found in the previous works on Mo and W^{10,11}.

- (5) Δdip is described in the end, but is not defined properly anywhere. This is hardly a canonical parameter that is well-known. The way it's calculated has to be clarified.

Answer: Many thanks for the suggestion. In the revised manuscript, the definition of Δdip is clearly described in the beginning. It is the difference in the Hartigen's dip test statistics between the LDOSs of the reference and defect atom. In addition, the calculation procedure of the Hartigen's dip test on LDOS is described in detail in the *method* section of the manuscript.

- (6) "Moreover, the Δdip parameter of atoms near the defects in pure W configurations can serve as a useful indicator for the solute-defect interaction energies and the site occupation preference of the transition metal solute without performing any calculations with substitutions of solute atoms." -> This test should be performed then, showing how well it performs (or did I misunderstand something here)?

Answer: The reviewer is right. The sentence was intended to mention that one can establish the correlation via a few of calculations on simple defect structures, and then use the established relationship to directly predict the interaction energy of each atomic site for other defect structures. Therefore, it is necessary to verify the prediction reliability of the established correlation relationship. To do so, we have performed the linear regression based on Eq. R3 for only a part of interaction energy data in the W-Re system, and then testify whether the obtained regression relationship can predict the rest interaction energies that were not included in the regression process. The results are shown as a parity plot in Fig. R4.

Figure R4. Model-predicted E_{int}^{fix} of the Re-vacancy and Re-dislocation interactions in W matrix in comparison with the results of the DFT calculations. The model is established based on Eq. R3 using the interaction energy data of the rest crystalline defects that studied in the present work (*exclude* the vacancy and dislocation defects).

As shown in Fig. R4, most of the predictions are in good agreement with the DFT results (RMSE \approx 0.025 eV). This indicates that one can use the correlation relationship established from the data of a few point and planar defects to estimate the interaction energy of each atomic site for the defects with higher computational complexity, such as the $\frac{1}{2}\langle 111 \rangle$ screw dislocation in the present work.

Corresponding revision has been made in the main text and the analyses of Fig. R4 are added in supplementary materials.

Results:

- (7) Normally, the correlation error (figs 4) that give an indicative interaction energy from the Δdip is way too large - This level of precision can hardly be used for something like alloy property screening (even though the correlation itself is quite interesting and persuasive).

Answer: Many thanks for the comment. We agree with the referee's opinion that there were a number of cases with relatively large fitting errors in the previously observed linear correlation between E_{int}^{fix} and Δdip . These errors mainly originate from the fact that the Δdip parameter can only describe the energy difference associated with the d -band filling, which is recognized as only a part of contribution to E_{int}^{fix} . As discussed above (see the reply to the comment #3 of Reviewer #1), in addition to the effects of d -band energy, the energy

contribution from the valence *sp*-band must be taken into account to quantitatively describe E_{int}^{fix} . Therefore, by incorporating both contributions from the valence *d*- and *sp*-band, the newly proposed regression model indeed achieve higher fitting accuracy for both W-Re and W-Pt systems as shown in Table R1. Especially, for the W-Re system, which is one of the most important alloy systems in practical applications, the current regression model could accurately capture E_{int}^{fix} in a wide range from -0.2 to 1.0 eV, with a standard error as small as 39 meV. Moreover, since the present model is simply based on a linear regression method, we expect the fitting accuracy can even be further improved by applying more advanced regression techniques in the future. In summary, with the improved prediction precision, we believe observed universality of the solute-defect interaction in the present work would be able to be directly used or at least provide useful insights for the practical alloy design in the future.

(8) In the correlation plots (Figs 4), the comparison of the delta(dip) and fixed interaction energies seem strongly biased to the 110-GSF results. This is a little worrisome, since there is no physical reason to have more such data points than anything else.

Answer: Many thanks for the comment. We agree with reviewer that this could artificially make the model overweight the data from the (110)-GSF calculations to the data from other defect calculations. Therefore, in order to avoid this issue, as shown in Fig. R3, instead of including all the data, only the interaction energies of the GSF along the <111> gliding direction are included for regression.

(9) Clearly the 112-GSF displays a different trend (very clear for Pt in W, still visible for Re in W). The authors don't comment this and it merits commenting.

Answer: Many thanks for pointing this issue out. One of the reasons that the (112)-GSF data is off-trend is the linear fitting in the previous manuscript artificially overweight the data from the (110)-GSF calculations, which results in an underestimation of the slope. As shown in Fig. S7 in the revised supplementary materials, the (112)-GSF data are no longer far off from the trend after including more data from other cases of solute-defect interactions in the regression process. In addition, as shown in Figs. R3b and R3c, after considering the contribution from the valence *sp*-band, E_{int}^{fix} of both the (112)- and (110)-GSFs in W-Re and W-Pt systems are now well described by the new regression model.

(10) Also, what can quite often be of very high significance in alloy design, is reliable information on solute-vacancy interactions, and the correlation for that defect with the delta(dip) seems tenuous at best.

Answer: Thanks for the comment. As shown in Fig. R3, after considering the energy contribution from the valence *sp*-electrons, the solute-vacancy interactions are described with smaller deviations by the newly proposed correlation model.

- (11) Why do the authors only compare the static interaction energies? The relaxation effects can be quite important (even if the authors show how large this effect is for one case). Since the authors omit to display the correlation (or lack of such) for the relaxed interaction energies with the delta(dip), one wonders as to the reason. A strong correlation with relaxed interaction energies would be more persuasive (and useful!).

Answer: Many thanks for the comment. First, it is worth to mention that the defect structures in our calculations were initially fully relaxed to reach its equilibrium state in pure matrix metals (W or Ta), and then used for solute substitution. The interaction energies were calculated under two conditions: forbidding and allowing atomic position relaxation during the total energy calculations of the solute-doped defect structures. Therefore, the difference between the relaxed (E_{int}^{relax}) and fixed-lattice interaction energies (E_{int}^{fix}) just gives the energy change due to the relaxation of the matrix metal structures upon the substitution of a solute atom. In addition, the fixed-lattice interaction energies (E_{int}^{fix}) themselves already contain a large portion of relaxation effects due to the defect structures in pure metals. As shown in Table S4 and S5, the differences between the relaxed (E_{int}^{relax}) and fixed-lattice (E_{int}^{fix}) interaction energies are small for many cases in both Ta- and W-based alloys. This indicates that the defect-solute interactions in bcc refractory metals may mainly originate from the changes in local electronic bonding near the defects sites rather than the energy change from the relaxation of the atomic positions upon the solute substitution. Therefore, in order to clearly reveal the quantitative correlation between the interaction energy and local electronic structure, we have used the fixed-lattice interaction energies to eliminate the minor effects of the lattice relaxation on the interaction energies. In addition, since the effects from the lattice relaxation is small, we expect the correlation is also generally valid for the relaxed interaction energies in most of the cases, especially for the defect site which have relatively strong interactions with the solutes.

Reference:

1. Hodges, L., Ehrenreich, H. & Lang, N. D. Interpolation scheme for band structure of noble and transition metals: ferromagnetism and neutron diffraction in Ni. *Physical Review* **152**, 505 (1966).
2. Mueller, F. M. Combined interpolation scheme for transition and noble metals. *Physical Review* **153**, 659 (1967).
3. Pettifor, D. G. Accurate resonance-parameter approach to transition-Metal band structure. *Physical Review B* **2**, 3031 (1970).
4. Pettifor, D. G. Theory of energy bands and related properties of 4d transition metals. III. s and d contributions to the equation of state. *Journal of Physics F: Metal Physics* **8**, 219 (1978).
5. Lambert, R. M. & Pacchioni, G. *Chemisorption and Reactivity on Supported Clusters and Thin Films: Towards an Understanding of Microscopic Processes in Catalysis*. **331**, (Springer Science & Business Media, 2013).
6. Xin, H., Holewinski, A., Schweitzer, N., Nikolla, E. & Linic, S. Electronic structure engineering in heterogeneous catalysis: Identifying novel alloy catalysts based on rapid screening for materials with desired electronic properties. *Topics in Catalysis* **55**, 376–390 (2012).
7. Harrison, W. A. *Electronic structure and the properties of solids: the physics of the chemical bond*. (Courier Corporation, 2012).
8. Ventelon, L., Willaime, F., Fu, C. C., Heran, M. & Ginoux, I. Ab initio investigation of radiation defects in tungsten: Structure of self-interstitials and specificity of divacancies compared to other bcc transition metals. *Journal of Nuclear Materials* **425**, 16–21 (2012).
9. Woodward, C. & Rao, S. I. Flexible Ab initio boundary conditions: simulating isolated dislocations in bcc Mo and Ta. *Physical review letters* **88**, 216402 (2002).
10. Trinkle, D. R. & Woodward, C. The chemistry of deformation: how solutes soften pure metals. *Science* **310**, 1665–1667 (2005).
11. Hu, Y.J. *et al.* Solute-induced solid-solution softening and hardening in bcc tungsten. *Acta Materialia* **141**, 304-316(2017).
12. Zhang, X. *et al.* Effects of solute size on solid-solution hardening in vanadium alloys: A first-principles calculation. *Scripta Materialia* **100**, 106–109 (2015).
13. Shi, S., Zhu, L., Zhang, H., Sun, Z. & Ahuja, R. Mapping the relationship among composition, stacking fault energy and ductility in Nb alloys: A first-principles study. *Acta Materialia* **144**, 853–861 (2018).
14. Steinhardt, P. J., Nelson, D. R. & Ronchetti, M. Bond-orientational order in liquids and glasses, *Physical Review B* **28**, 784 (1983).
15. Gombert, J. A., Medford, A. J. & Kalidindi, S. R. Extracting knowledge from molecular mechanics simulations of grain boundaries using machine learning. *Acta Materialia* **133**, 100–108 (2017).
16. Mueller, T., Kusne, A. G. & Ramprasad, R. Machine learning in materials science:

- Recent progress and emerging applications. *Reviews in Computational Chemistry* **29**, 186–273 (2016).
17. Dezerald, L., Proville, L., Ventelon, L., Willaime, F. & Rodney, D. First-principles prediction of kink-pair activation enthalpy on screw dislocations in bcc transition metals: V, Nb, Ta, Mo, W, and Fe. *Physical Review B* **91**, 94105 (2015).
 18. Han, J., Thomas, S. L. & Srolovitz, D. J. Grain-Boundary Kinetics: A Unified Approach. *Progress in Materials Science* **98**, 386-476 (2018).
 19. Yasi, J. A., Hector, L. G. & Trinkle, D. R. Prediction of thermal cross-slip stress in magnesium alloys from direct first-principles data. *Acta Materialia* **59**, 5652–5660 (2011).
 20. Leyson, G. P. M., Curtin, W. A., Hector, L. G. & Woodward, C. F. Quantitative prediction of solute strengthening in aluminium alloys. *Nature Materials* **9**, 750 (2010).
 21. Itakura, M., Kaburaki, H., Yamaguchi, M. & Okita, T. The effect of hydrogen atoms on the screw dislocation mobility in bcc iron : A first-principles study. *Acta Materialia* **61**, 6857–6867 (2013).
 22. Tsuru, T. & Suzudo, T. First-principles calculations of interaction between solutes and dislocations in tungsten. *Nuclear Materials and Energy* **16**, 221–225 (2018).

Reviewers' comments:

Reviewer #3 (Remarks to the Author):

Review of resubmitted manuscript:

The authors have now updated their study after quite similar criticisms by three referees. In general, their responses are satisfactory and proper measures have been implemented. The manuscript is now much more convincing and is closer to an acceptance recommendation. Many questions were resolved but a few remain or appeared in this iteration:

1) There is no information on how sensitive the delta dip calculation is to the parameters used to generate the LDOS. Now the authors use 301 energy bins. This does first of all not give a well-defined physical energy binning since the initial and final energies may be different. The authors should state the interval as well, or better yet, state the range of each energy bin. Furthermore, what happens if 100, 500, 1000, 2000 bins are used? How sensitive is the delta dip to this choice. How sensitive is it to other parameters (k-points, cutoff, etc). The authors must perform studies of this and add this information to the supplementary materials for me to be satisfied. The description of the dip test is still not sufficiently detailed for me. Show us the central equations.

2) The use of a p-value is not something I would recommend in this setting. The p-value, and it's generic message is heavily criticized in applied statistics literature. Use only the standard measures instead.

3) One important issue the authors fail to mention is that even though this method seems to give good general results, it is not always sufficiently accurate to be really useful as a screening tool. In determining important properties of a material under stress of some kind (generically speaking), it is often not the general interaction energies that are important, but certain low energy configurations - and their exact values can matter a lot. For example, a negative binding energy that becomes positive, if it's for the most important defect couple, changes completely the physics. With this said, I think the authors need to downplay a bit the stated importance of the here developed method. Their conclusions will most likely not hold in reality just because what is most important is not to describe average property behavior well, but to describe particular properties very well. Still, I think this method is very interesting and opens up for some deeper analysis of the underlying physics behind solute-defect interactions.

4) The authors should report how well the method works when applying the canonical four supercell method for calculating binding energies instead of the one here applied with two separated objects in the same cell, far removed.

5) In figure(s) 3, the full two-component method is not applied, but rather the old method containing just the delta dip. Why is this done? Since a new, better method is now developed, should not this be showcased?

6) The outliers in figures 4 and 5 need to be mentioned and discussed. Can reasons be found for all glaring cases?

7) Typo, in the description of the dip test: "which measures the deviant" should read "which measures the deviation"

If the authors take proper care of adjusting their manuscript, I will recommend publication after this major revision.

Reviewer #4 (Remarks to the Author): I will confine my comments to assessing whether or not the authors have adequately addressed the major comments of the previous reviewers, and the reviewers' assessment that the work lacks sufficient breadth to justify the claimed universality. The goal of the paper is laudable – can the LDOS of sites around a defect provide insight into the binding energy of solutes to the defect, in the absence of full direct electronic structure calculations? This kind of thinking has long driven the understanding of relative crystal structure stability, for instance. All the reviewers appreciate the goal of the paper, and the progress made by the authors. In response to the reviewers' requests for more generality and accuracy, the authors have now demonstrated a broad correlation between two features, the Δ_{dip} (the focus of the original paper) and an x_{sp} factor, and included additional solute binding energies to a wide range of defects in several refractory matrices. The need to extend the correlation beyond the Δ_{dip} reflects the lack of predictability of the original model. The inclusion of x_{sp} , attempting to capture sp-d binding, is a good step forward in improving agreement but the final result is not very satisfying to this reviewer, and I suspect to the original reviewers. The issues are that (i) the correlation is now based on fitting of some arbitrary coefficients a_1 and a_2 , and (ii) the level of agreement still remains moderate at best.

- 1) Regarding the coefficients, a universal concept would show that the coefficients a_1 and a_2 are (nearly) constant such that the “material” and “solute” information is buried in the physical parameters Δ_{dip} and x_{sp} . Unfortunately, this is not the case. The a_1 and a_2 values vary widely, and the a_2 value changes sign – argued to be related to band filling of the solutes, which is fine but highlights that there is more physics than the current correlation can capture.
- 2) Regarding the level of agreement, the authors show an overall RSME error across many defects for a given matrix/solute pair. The RSME error is small and the p-value is very small, both positive factors. But the actual predictive capability of the method is probably not sufficient for practical use. Calculations of segregation concentrations of solutes to grain boundaries at a given temperature would not be generally predicted with sufficient accuracy using this approach (assuming for the moment that the DFT value is accurate and that relaxation effects are small). The authors do not test real accuracy. That is, given a new grain boundary structure (or some other structure) not previously examined, what do they predict for solute binding energies and what is computed in DFT? The deviations (absolute and percentage-wise especially for low energy defects) would seem to be too large to be useful.
- 3) Stepping back from the search for universal correlating factors, it is probably important to recognize that many of the quantities studied here can now be computed relatively easily with direct DFT, and with full relaxation. The emergence of high-throughput methods, even if exchanging accuracy for speed by reducing convergence and accuracy of the DFT, has emerged as an approach that can cover a very wide space. The computations of solute interactions with point defects is probably particularly easy, and with grain boundaries is moderately efficient although not for very complex boundaries. Computations of solute/dislocation interaction energies are probably the most challenging due to the system sizes required, but it is also here where the accuracy of the present method may again be insufficient. That is, the interaction energies may be small but the differences between the correlation and the DFT are likely to large enough to be important, while not including

relaxations that presumably become more important as the absolute energies become small. The authors might wish to claim that the correlation is meant precisely for issues such as complex grain boundaries that are computational intensive, but they have not established what level of accuracy is really required for physical problems of interest and whether their methods would be useful for alloy design/selection (as noted by one of the reviewers).

- 4) The authors are to be commended for extensive and detailed replies to the reviewers, extension of their ideas to include sp-d effects in some way and therefore achieve better correlation (albeit with fitting parameters), and for the overall exploration of a promising and potentially very valuable correlation. However, the actual achievement still falls somewhat short of the goals. The proposed universal correlation is not sufficiently universal nor sufficiently accurate. I wish it were otherwise, and that the authors could have achieved their goals. Certainly the field of materials science is filled with rather weaker correlations that have been widely adopted by experimentalists and pursued by modelers due to the ease of computation, but those trends are unfortunate because they are also not accurate and not sufficiently predictive. I believe that the revised paper could/should be published in an archival journal so that the ideas are disseminated and evaluated, and the community will learn from this effort. But publication in Nature Communications is probably not warranted because the limitations of the work make it of much narrower and less definitive value than is usually sought by the editors of Nature Communications.

Reviewers' comments:

Reviewer #3 (Remarks to the Author):

Review of resubmitted manuscript:

The authors have now updated their study after quite similar criticisms by three referees. In general, their responses are satisfactory and proper measures have been implemented. The manuscript is now much more convincing and is closer to an acceptance recommendation. Many questions were resolved but a few remain or appeared in this iteration:

Answers: Thanks for the reviewer's appreciations and valuable suggestions. Here, we make point-to-point replies to answer the reviewer's questions and comments, especially the detailed analyses of certain "outlier" data as pointed by the reviewer. Intrigued by these outlier analyses and several recent studies on electronic structure effects to properties of transition metals/alloys, we add a high-order statistical term on the basis of our original linear model to further reduce the residuals between the model predictions and the direct DFT calculations. Especially, the large errors of the outliers from the original linear model are significantly reduced. We also provided detailed examples to apply our correlation model to predict solute-defect binding energies and solute segregation concentrations for new complex grain boundary structures that are not in the training dataset of the model. The results suggest our model would be good enough for approximately describing the defect properties that are overall determined by the solute-defect interaction energies of multiple defect sites. On the other hand, for the defect properties that are only determined by one or several critical sites, the present model may have the risk to yield relatively large errors and needs to be further improved in our future research by including more DFT data.

We have added the above discussions of the potential capability and the limitation of our model in the revised manuscript. Finally, we rephrased the "**electronic factors**" to "**electronic descriptors**" in this reply and the revised manuscript (all changes highlighted in blue color), and we change the title of our manuscript to "**Local electronic descriptors for solute-defect interactions in bcc refractory metals**". We think this paper is improved a lot thanks for the reviewer's comments and is ready to be published in *Nature Communications*.

1) There is no information on how sensitive the delta dip calculation is to the parameters used to generate the LDOS. Now the authors use 301 energy bins. This does first of all not give a well-defined physical energy binning since the initial and final energies may be different. The authors should state the interval as well, or better yet, state the range of each energy bin. Furthermore, what happens if 100, 500, 1000, 2000 bins are used? How sensitive is the delta dip to this choice. How sensitive is it to other parameters (k-points, cutoff, etc). The authors must perform studies of this and add this information to the supplementary materials for me to be satisfied.

Answer: Thanks for the comments and suggestions. In the revised manuscript, the interval value of the energy bin was reported. Since we used the default settings in VASP for determining the minimum/maximum energy boundaries of the LDOS, the interval of each individual LDOS calculation is slightly varied, ranging from 0.151 to 0.155 eV.

Furthermore, the sensitivities of the Δdip measurements to the LDOS-related DFT parameters (i.e. number of bins, k-points density, cutoff energy and width of smearing) were tested. It is found that convergences of the Δdip measurements can be quickly achieved by increasing the number of the bins and density of the k-points meshing. The settings used in the present calculations are within the convergence range. In addition, it is found that the Δdip measurements are insensitive to the cutoff energy for the plane-wave basis and the width of smearing. More details of the test results are described and discussed in **Section 4** in the revised Supplementary Information.

2) The description of the dip test is still not sufficiently detailed for me. Show us the central equations.

Answer: Thanks for the suggestion. The Hartigan's dip test is described in more detail in the methodology section of the revised manuscript.

3) The use of a p-value is not something I would recommend in this setting. The p-value, and its generic message is heavily criticized in applied statistics literature. Use only the standard measures instead.

Answer: Thank you very much for pointing this out. The p-value of the linear regression and corresponding discussion were removed from the revised manuscript.

4) One important issue the authors fail to mention is that even though this method seems to give good general results, it is not always sufficiently accurate to be really useful as a screening tool. In determining important properties of a material under stress of some kind (generically speaking), it is often not the general interaction energies that are important, but certain low energy configurations - and their exact values can matter a lot. For example, a negative binding energy that becomes positive, if it's for the most important defect couple, changes completely the physics. With this said, I think the authors need to downplay a bit the stated importance of the here developed method. Their conclusions will most likely not hold in reality just because what is most important is not to describe average property behavior well, but to describe particular properties very well. Still, I think this method is very interesting and opens up for some deeper analysis of the underlying physics behind solute-defect interactions.

Answer: Many thanks for the advice and we agree with the reviewer. In the revision, we have rephrased the corresponding statements in the introduction, discussion and conclusion parts. We hope now our expressions on the importance of the work is more accurate and appropriate.

We also totally agree with the reviewer that a deeper analysis of the underlying physics is desired in the future to establish more accurate prediction models, especially for the defect sites that critical for the physical properties of defects. In this round of revision, we further improved the prediction accuracy by adding a residual-correction function that based on multiple electronic descriptors to our original linear correlation model. Especially, the large errors of the outliers from the original linear model are significantly reduced. The results are summarized and discussed in Figs. 5 and 6 in the main text and Fig. S17 in the revised Supplementary Information. It shows that the predictions from the revised model would be accurate for describing the properties that overall determined by the interaction energies of multiple defect sites, such as the isotherms of the solute GB segregation concentration. The model predictions are verified with two new complex grain boundary structures that are not included in the original regression dataset. We also agree with the reviewer that the accuracy of our model should be further improved for the prediction of defect properties that are determined by solute interactions with one or several particular defect sites. Meanwhile, since the residual corrections are established based on a statistical method and careful cross-validations, their accuracy can be further improved if more DFT training data can be included in future research. To achieve this purpose, we will share all of our data on open-access database and keep performing more defect calculations in the future. As suggested by the reviewer, we have added these discussions of the potential capability and the limitation of our model in the revised manuscript.

5) The authors should report how well the method works when applying the canonical four supercell method for calculating binding energies instead of the one here applied with two separated objects in the same cell, far removed.

Figure R1. Comparison between the E_{int}^{fix} calculated from four-supercell method and predicted from the regression model.

Answer: Thanks for the comment. Taking the W-Re system as an example, a corresponding investigation was performed for the interaction energies that were calculated using the four-supercell method^{1,2}. A 3×3×3 bulk bcc supercell is used as the reference configuration. The solute-defect interaction is then calculated as, $E_{int} = (E_{def}^W + E_{ref}^X) - (E_{def}^X + E_{ref}^W)$, where E_{def}^X (respectively E_{def}^W) is the total energy of the defect supercell containing (respectively not containing) the solute atom and E_{ref}^X (respectively E_{ref}^W) is the energy of the bulk bcc supercell with (respectively without) the solute atom. Correspondingly, Δdip and x_{sp} of the defect atoms are also recalibrated using a W atom in the bulk bcc supercell as the reference, instead of using a W atom in the defect-containing supercell as the reference in the two-supercell method. The regression result is shown by a parity plot in Fig. R1 above. It is important to note that the four-supercell calculation is not performed for the $\frac{1}{2}\langle 111 \rangle$ screw dislocation due to the limitation of the flexible boundary condition method specifically used for dislocation core calculations. As shown in Fig. R1, the interaction energies calculated by the four-supercell method can still be well described by the linear correlation proposed in the present work. The regression coefficients of the Δdip and x_{sp} parameters are 14.21 eV and -1.43 eV, respectively, which are slightly different from the values obtained by the two-supercell method (15.98 eV and -1.29 eV, respectively). This is because different reference states are used to calculate the interaction energies and electronic descriptors in the two methods. On the one hand, using the two-supercell method, both of the defect and reference atoms are in a same supercell containing the defect structure. As a result, the defect-induced and supercell-size-induced strain effects are largely excluded in the calculations of the interaction energy and Δdip and x_{sp} parameters. On the other hand, the contribution of these strain effects is included in the four-supercell calculations as the reference atom is now in a perfect bcc lattice supercell with relatively large supercell size. Consequently, the regression coefficients of the four-supercell calculations are slightly different from those obtained by two-supercell method. The above discussion is also added in revised Supplementary Information.

6) In figure(s) 3, the full two-component method is not applied, but rather the old method containing just the delta dip. Why is this done? Since a new, better method is now developed, should not this be showcased?

Answer: Thanks for the comment. We agree with the reviewer that the current form of Fig. 3 is somehow confusing. In fact, by organizing Fig.3 that way, we were intending to show that the solute-defect interaction energy is strongly correlated with Δdip of the matrix atom at the corresponding substitution site, and in the meanwhile, only using Δdip parameter is not enough to capture the interaction energy quantitatively in general cases. In the revised manuscript, we reduced Fig. 3 (Fig.2 in the revised manuscript) into two sub-figures to show this intention more clearly.

7) The outliers in figures 4 and 5 need to be mentioned and discussed. Can reasons be found for all glaring cases?

Answer: Many thanks for the comment. After further analyses, we found that the prediction outliers usually appear to be some particular defect sites, independent of the types of the solute elements. For example, the present linear model yields relatively large deviations when predicting the interaction energies of the *site 3* of the (111)-GB (Fig. S3b) in both the W-Ta (an overestimate of 0.120 eV) and W-Re systems (an underestimate of 0.144 eV). Additionally, the solute-interactions of the <100>-dumbbell defect seem to be poorly described in the W-Pt, Ta-Hf and Ta-Os systems, especially the first- and second-nearest sites to the defect center. Further scrutinizing the electronic structures of the outlier defect atoms, it is found that the variations in their LDOSs relative to the reference atom may not be sufficiently characterized by the Δdip and χ_{sp} parameters, which correspondingly results in large errors in predictions of the interaction energy. Details are described in the two following paragraphs.

In Fig. R2a the *d*-orbital LDOS for a W atom that occupying the *site 3* of the (111)-GB (refer as the outlier defect atom) is plotted to compare with the LDOSs of the reference W atom far away from the defect and a W atom at a site near the (210)-GB (referred as the normal defect atom, where the present model yields an accurate prediction on the interaction energy). As illustrated by the shapes of the whole LDOSs curves, both the outlier and normal defect atoms have less bimodal LDOSs compared to the reference atom. Especially, their LDOSs are very close to each other in the region around the bcc pseudo-bandgap. This implies that the contributions of the *d*-band filling effect to the interaction energy should be quite similar for these two atoms when they are substituted by the Ta or Re solutes. Therefore, one would expect the outlier and normal atoms to have a similar value of the Δdip parameter to correctly reflect the *d*-band filling effect. However, in practice, due to the presence of an abnormally higher peak in right side band of the outlier atom, its Δdip is calculated to be 0.0109, much smaller than that of the normal atom, 0.0197. Correspondingly, the interaction energy of the outlier site is overestimated by the linear model for Ta-substitution and underestimated for Re-substitution. This inconsistency suggests that more descriptors of LDOSs should be included to fully describe their effects. For example, more information can be obtained if the LDOSs of the outlier and normal defect atoms are plotted into the t_{2g} and e_g orbital sets individually. As shown in Fig. R2b, the abnormally high peak of the outlier atom is mainly originated from the e_g orbital set. Additionally, it can be seen that the LDOS of the t_{2g} set of the two atoms are quite close to each other, which yields a similar value of Δdip if we only perform the measurements on the t_{2g} set. These results suggest that the bimodality variations of the individual t_{2g} and e_g orbital sets may serve as additional descriptors to further reduce the prediction errors for the outlier site.

Figure R2. (a) the d -orbital LDOS for a W atom that occupies the *outlier defect site* of the (111)-GB along with the LDOSs of the reference W atom and a W atom at a *normal defect site* of the (210)-GB where the present model yields a normal prediction on the interaction energy. (b) The LDOSs of the t_{2g} and e_g orbital sets of the outlier and normal W atoms in (a). (c) the d -orbital LDOS for a W atom that occupies the *outlier defect site* of the <100>-dumbbell along with the LDOSs of the reference W atom and a W atom at a *normal defect site* where the present model yields an accurate prediction on the interaction energy.

For another set of outliers, the first- and second nearest sites of the <100>-dumbbell defect, we also found that there may be additional features of their LDOSs that are not adequately described by the Δdip and χ_{sp} parameters. Similar to Fig. R2a, in Fig. R2c we plotted the d -orbital LDOS for a W atom that occupies the first-nearest site of the <100>-dumbbell defect along with the LDOSs of the reference W atom and a W atom at a normal defect site, where the linear model yields an accurate prediction on the interaction energy. In addition, it is worth to mention that the outlier and normal defect atoms have similar values of the Δdip parameters (~ 0.0301). On the one hand, the LDOS of the outlier atom has a less bimodal shape relative to the LDOS of the reference atom, which is indeed reflected by a positive value of Δdip . On the other hand, the LDOS shapes of the outlier and normal defect atoms are quite different from each other although their Δdip have similar values. As shown in Fig. R2c, close to the left band edge, there are two sharp peaks in the LDOS of the outlier defect atom, which are not seen in the LDOS of the normal defect atom. Consequently, the effects of these peaks on the solute-defect interaction energy may not be adequately captured by the Δdip measurement, correspondingly resulting in a large prediction error. In addition, another possible reason for the <100>-dumbbell outliers could be the approximation on using the Voronoi volume to represent the average interatomic distance for constructing the χ_{sp} parameter. This average may have large uncertainties in terms of the standard deviation when defect-induced lattice distortions become large and anisotropic like the <100>-dumbbell defect.

The above discussion of Fig. R2 suggests that further investigations may be needed to discover additional electronic descriptors to yield more thorough descriptions of the defect LDOSs and incorporate them into more accurate prediction models. Therefore, in this round of revision, we have attempted to further improve the accuracy of our model by including additional information

of the defect electronic structures. Specifically, on the basis of the linear model, a new function established by statistical learning is added to correct the residuals of the linear regression. In light of the above discussion, the statistical learning is performed based on a broader set of electronic descriptors, such as bimodality variations of the individual t_{2g} and e_g orbital sets, the band center and root-mean-square width of the sp - and d -orbitals and so on. Several recent DFT studies suggest that these electronic descriptors could also be related to the energetic stability of transition metals/alloys³⁻⁷. In addition, to further validate our modified model, the solute-defect interaction energies of two new complex grain boundary structures were calculated and used as a test set. The prediction results based on this revised model suggest the errors of the outliers can be significantly reduced. These new findings are summarized and discussed in Figs. 5 and 6 in the main text and Fig. S17 in the revised Supplementary Information. The above analysis on outliers is also added as **Section 12** in the revised Supplementary Information.

In addition, it would be natural to search the atomistic structure features that determine whether such outliers of electronic structures occur. Although we have done systematic investigations, so far we haven't found clear and comprehensive mechanisms on the difference between atomistic structures of the normal defect sites and those of the outlier defect sites (except some particular case like too short bond length in $\langle 100 \rangle$ -dumbbell). It suggests two conclusions. First, the correlation between atomistic structures and electronic structures are too complex (determined by quantum mechanics), so it is very difficult to find some simple and quantitative correlation from a limited number of data points. It is more likely to find such correlations based on large size of datasets and statistical learning, which will be our next-step task. Second, with the limited number of data points as this work, the electronic descriptors listed above are indeed effective for describing the energetics of transition metals/alloys with different atomistic structures, suggesting these descriptors have high representability in complex compositional and structural parameter spaces.

8) Typo, in the description of the dip test: "which measures the deviant" should read "which measures the deviation"

Answer: Thanks for pointing this out. The typo is corrected in the revised manuscript. We also did an additional proofreading one more time to avoid other typos.

9) If the authors take proper care of adjusting their manuscript, I will recommend publication after this major revision.

Answer: We thank the reviewer very much for the thorough reading of our manuscript and many insightful comments. We believe the manuscript is now stronger for the changes, and hope that it is qualified for publication in *Nature Communications*.

Reviewer 4

Review of Nat Comm 164358

I will confine my comments to assessing whether or not the authors have adequately addressed the major comments of the previous reviewers, and the reviewers' assessment that the work lacks sufficient breadth to justify the claimed universality. The goal of the paper is laudable – can the LDOS of sites around a defect provide insight into the binding energy of solutes to the defect, in the absence of full direct electronic structure calculations? This kind of thinking has long driven the understanding of relative crystal structure stability, for instance. All the reviewers appreciate the goal of the paper, and the progress made by the authors. In response to the reviewers' requests for more generality and accuracy, the authors have now demonstrated a broad correlation between two features, the Δ_{dip} (the focus of the original paper) and an x_{sp} factor, and included additional solute binding energies to a wide range of defects in several refractory matrices. The need to extend the correlation beyond the Δ_{dip} reflects the lack of predictability of the original model. The inclusion of x_{sp} , attempting to capture sp-d binding, is a good step forward in improving agreement but the final result is not very satisfying to this reviewer, and I suspect to the original reviewers. The issues are that (i) the correlation is now based on fitting of some arbitrary coefficients a_1 and a_2 , and (ii) the level of agreement still remains moderate at best.

Answer: Many thanks for the reviewer's help to review the comments from other reviewers and give valuable comments to help us further improve our manuscript. We address the reviewer's comments and concerns point-by-point as the following. Here we summarize our revisions in advance.

First, we explain the exact meaning of the “universal” linear relation that we claimed in our last manuscript. The linear model is developed based on the moment and band theory from Pettifor⁸⁻¹² and the coefficient a_1 and a_2 are physically interpreted with the fundamental electronic properties of the solute elements (Detailed discussions in our Reply 1 to the reviewer's questions on the next page). To reduce the possible misunderstanding, we change the title of our manuscript to “Local electronic descriptors for solute-defect interactions in bcc refractory metals”, and we change the “universal correlation” to “general correlation” in the manuscript text. The generality of this linear model has been further confirmed by our prediction examples mentioned in the 3rd point below. We also rephrased the “**electronic factors**” to “**electronic descriptors**” in this reply and the revised manuscript.

Second, we improve the accuracy of our original linear model by using additional electronic descriptors and statistical methods. The general philosophy is to use the **physics-inspired linear model** to keep high **transferability** to describe various solute-defect interaction cases and **advanced statistical models** based on the local electronic descriptors to reduce the residuals of the linear model. The fitting accuracy of defect-solute interactions is indeed improved for all the five solute-matrix systems studied in the present work.

Third, we use new and complex grain boundaries (GBs) to show the **good predictability and low computational cost** of both our original linear model and the revised linear + residual-

correction model, where DFT data of these GBs **were not used for the regression and statistical training**. It shows that the model predictions would be good enough for approximately describing the properties that are determined by the interaction energies of multiple defect sites, such as the isotherms of the solute GB segregation concentration. In addition, the linear + residual-correction statistical model indeed has increased prediction accuracy. We acknowledge that the accuracy of our model should be further improved for the prediction of defect properties determined by solute interactions with one or several particular defect sites, which could be achieved if more DFT training data are included for the statistical model. Thus, as suggested by the reviewer, we will share all our data on an open-access database for the community to further enhance the accuracy and transferability by keeping generating new train dataset for different defect sites.

In summary, we have added the above discussions of the potential capability and the limitation of our model in the revised manuscript (**all changes highlighted in blue color**). We think this paper is improved a lot thanks for the reviewer's comments and is ready to be published in *Nature Communications*.

1) Regarding the coefficients, a universal concept would show that the coefficients a_1 and a_2 are (nearly) constant such that the “material” and “solute” information is buried in the physical parameters Δdip and x_{sp} . Unfortunately, this is not the case. The a_1 and a_2 values vary widely, and the a_2 value changes sign – argued to be related to band filling of the solutes, which is fine but highlights that there is more physics than the current correlation can capture.

Answer: Many thanks for the comment. We agree with the reviewer that using the word “universal” may not be a very precise phrasing for the general correlation found in the present work. In the manuscript, we intended to show that the solute-defect interaction energies in binary alloys of bcc refractory metals (such as W and Ta) with transition-metal substitutional solutes are strongly correlated with two electronic descriptors, Δdip and x_{sp} , which can be calculated by just having the electronic information of the pure metal defects. There are two levels of generality for this correlation relation. On the first level, for all the tested solute-matrix pairs, their interaction energies can be generally described using the same linear formalism (Eq. 1 in the main text), while the values of the formalism parameters are determined according to the matrix materials and solutes. On the second level, the features and information of the defect structures and matrix elements are all generalized and integrated to the Δdip and x_{sp} parameters, which means the regressions of the W-Ta, W-Re and W-Pt systems are based on same sets of the Δdip and x_{sp} parameters with same values, and so as for the Ta-Hf and Ta-Os systems. The material-dependent features are reflected by the fitting coefficient a_1 and a_2 , which can be physically interpreted with the electronic properties of the solute elements. a_1 is correlated to the solute-induced shifting in the relative position of the Fermi level on the LDOS compared to that of the matrix element, as described in detail in the second paragraph of the **Discussion** section in the revised manuscript (Page 20 of the revised manuscript). a_2 is understood in terms of the difference in the spatial extent of d -orbital between the solute and matrix elements, as shown by the derivations of Eq. S10-S12 in **Section 16** of the Supplementary Information.

As a comparison, our general correlation is similar to the well-known d -band center model in the heterogeneous catalysis field¹³: the adsorption energies of certain atom/molecule on transition metal/alloy surface may linearly vary according to the d -band center of surfaces with different structures and compositions, and the linear coefficient depends on the types of adsorbed atom/molecule. In the revision, we have made the corresponding changes to clearly express both the generality and limitations of the present model. In addition, we used the words “general/generality” to replace “universal/universality” in the revised manuscript, which we believe providing a more precise description of the major conclusions of the present work.

2) Regarding the level of agreement, the authors show an overall RSME error across many defects for a given matrix/solute pair. The RSME error is small and the p-value is very small, both positive factors. But the actual predictive capability of the method is probably not sufficient for practical use. Calculations of segregation concentrations of solutes to grain boundaries at a given temperature would not be generally predicted with sufficient accuracy using this approach (assuming for the moment that the DFT value is accurate and that relaxation effects are small). The authors do not test real accuracy. That is, given a new grain boundary structure (or some other structure) not previously examined, what do they predict for solute binding energies and what is computed in DFT? The deviations (absolute and percentage-wise especially for low energy defects) would seem to be too large to be useful.

Answer: We thank the reviewer for the helpful comments and advice. Correspondingly, there are three major revisions: (i) we improved the regression accuracy of the linear model through adding a residual-correction function that established by statistical learning of a broader set of electronic descriptors; (ii) To further validate this improved model, the solute-defect interaction energies for multiple defect sites in two new complex GBs (i.e. $\Sigma 13$ (230)-GB and $\Sigma 27$ (552)-GB, both of which are described in large supercells containing more than 200 atoms) were calculated and used as a test set, which means DFT data of these GBs were not used for the regression and statistical training; (iii) To examine the prediction accuracy of the model to certain defect properties, the solute segregation concentrations of the two tested GBs are calculated using the model-predicted interaction energies based on the White-Coghlan site occupation model^{14,15}, and compared with the those obtained using the DFT-computed interaction energies. More details are described as follows.

First, we show that the prediction accuracy can be further improved by combining the linear regression model with a residual-correction function that is established by statistical learning of a broader set of electronic descriptors.

In the present work, we introduced two electronic descriptors, Δdip and x_{sp} , to quantitatively generalize the local variations in the electronic structures near the defect. The relatively good regression qualities (e.g. small RMSE and close-to-one R^2 values) strongly suggested that these two descriptors are indeed robust to capture the major energy contributions to the solute-defect bindings through a linear relation. On the other hand, some recent DFT calculations also indicate that the various energetic properties of the transition metal alloys could connect closely with many other local features of the d -band, such as transition between e_g and t_{2g} orbital sets⁷, e_g/t_{2g}

population ratio³, occupation fraction^{5,6}, upper band edge⁴ and so on. This makes us consider that the remaining residuals of the present linear model may be further described by other local features of the electronic band in addition to the d -band bimodality and sp - d hybridization.

Here, as a first approximation, we started with a set of potential descriptors (D_i, D_j, \dots) that constructed based on some common features of the electronic bands, including the band center and root-mean-square width of the whole d -orbital, e_g and t_{2g} orbital sets, and the sp -orbitals, and individual bimodalities of the e_g and t_{2g} orbital sets. Similar to Δdip and x_{sp} , all these descriptors are also obtained from the DFT calculations of the defects in pure metals for each individual site. A detailed description of the procedures of the descriptor construction is included in **Section 13** in the Supplementary Information. Accordingly, we now propose the solute-defect interaction energy (E_{int}^{fix}) in the binary alloys of bcc refractory metal and transition metal solute can be approximated as,

$$E_{int}^{fix} \approx a_1 \Delta dip + a_2 x_{sp} + f_{r-c}(D_i, D_j, \dots) \quad \text{Eq. R1}$$

where the first two parts of the equation correspond to the linear model described by Eq. 1 with the same a_1/a_2 from Table 1 in the main text. $f_{r-c}(D_i, D_j, \dots)$ is a residual-correction function that is established based on the above-mentioned set of potential descriptors (D_i, D_j, \dots) by regressing the residuals (Δ_{linear}) of the linear model, where $\Delta_{linear} \equiv E_{int}^{fix}(DFT) - (a_1 \Delta dip + a_2 x_{sp})$.

We first performed covariance analyses between the regression residuals of the linear model and each of the potential descriptors to search whether there are strong and global correlations. Unfortunately, the result of the covariance analysis is negative. Therefore, instead of building a global function with analytical expression (e.g. another linear model), we attempt to use local regression methods to construct the residual-correction function based on the above set of potential descriptors. Here the local regression methods mean that the residual-correction function behaves differently depending on the detailed values of the potential electronic descriptors (D_i, D_j, \dots), so this ‘‘local’’ feature is appropriate to deal with some outliers for the linear model prediction due to some abnormal electronic features as discussed in the reply to Reviewer #3 (Reply 7 to Reviewer #3).

Specifically, a statistical model is proposed and developed based on multivariate local linear regression implemented in the **Locfit package**^{16–19}. We limited the variable dimension to be no more than four, meaning that the regression is only run with four descriptors or less at a time. This is because there is no significant improvement on the regression accuracy for the model with more than four descriptors, which could also increase the risk of overfitting. Within a cross-validation framework, the developed statistical model: (i) searches the descriptors that provide best regression accuracy on average in all the five solute-matrix systems studied in the present work; (ii) establishes a residual-correction function by recording the corresponding regression models. More details on the algorithms and calculation procedures of this statistical model are explained in **Section 13** the Supplementary Information.

The parity plots of regression results of the present improved model based on Eq. R1 (referred as the linear+ f_{r-c} model in the following) are shown in Figs. R3a and R3b for the W-Re and Ta-Hf

systems, respectively. As a comparison, the results solely obtained from the original linear model are also included in Fig. R3. As shown in both figures, on the basis of the original linear model, the newly developed Linear+ f_{r-c} model indeed yields better agreements with the original DFT results by adding the residual-correction function, $f_{r-c}(D_i, D_j, \dots)$. The parity plots of the W-Ta, W-Pt and Ta-Os systems are shown in Fig. S17 in the Supplementary Information, where visible improvements on regression accuracy are also observed.

Figure R3. E_{int}^{fix} from the linear+ f_{r-c} model based on Eq. R1. are plotted against those from direct DFT calculations. As a comparison, the results solely obtained from the original linear model are also plotted. (a) W-Re system; (b) Ta-Hf system.

As a quantitative comparison, the regression root-mean-squared errors (RMSE) of the linear+ f_{r-c} model and original linear model are listed in Table R1 below.

Table R1. The regression root-mean-squared errors (RMSE) of the present linear+ f_{r-c} model and original linear model for the original training data set of ~ 350 solute-defect interaction cases with 5 matrix-solute element pairs.

	Regression root-mean-squared errors (eV)				
	W-Ta	W-Re	W-Pt	Ta-Hf	Ta-Os
Linear	0.044	0.039	0.168	0.043	0.105
Linear+ f_{r-c}	0.022	0.024	0.080	0.021	0.072

Second, we performed additional DFT calculations on two new complex GBs, namely $\Sigma 13$ (230)-GB and $\Sigma 27$ (552)-GB, to study their electronic structures and solute-defect interaction energies in all the five solute-matrix systems. These new data are served as a test set, **meaning they are not included in the dataset for regression**, to validate the predictions from the linear+ f_{r-c} and our original linear model.

The sizes of the supercells used for DFT calculations in the bcc index are $[\bar{3}20] \times 2[230] \times 2[001]$ (208 atoms) and $2[1\bar{1}0] \times \frac{1}{4}[\bar{1}\bar{1}5] \times 5.61[552]$ (with 15 Å vacuum, 237 atoms) for the $\Sigma 13$ (230)-GB and $\Sigma 27$ (552)-GB, respectively. It is noteworthy that the input geometry of the $\Sigma 27$ (552)-GB is not constructed using the coincidence site lattice (CSL) theory but implemented from a ground state structure in W predicted by a state-of-art evolutionary structure search algorithm^{20,21}. Since this GB structure does not have the minor symmetry with the GB as the mirror plane, 15 Å of vacuum was applied along the GB plane normal direction to prevent the formation of two different GB structures in one supercell. The atomistic configurations of the W supercells after relaxation are shown in Figs. R4a and R4b for the $\Sigma 13$ (230)-GB and $\Sigma 27$ (552)-GB, respectively. The atomic sites chosen for the solute substitution are marked in red while the reference site is marked in blue. Because the corresponding calculations are quite computational consuming, the solute-defect interaction energies are only calculated for partial defect sites. So 11 sites for the $\Sigma 13$ (230)-GB and 15 sites for the $\Sigma 27$ (552)-GB were used for all the five solute-matrix element pairs investigated here. More sites that are very close to the GB-plane were also included for the specific W -Pt pair in the $\Sigma 27$ (552)-GB case, and the results show no noticeable differences in the prediction accuracy of the solute segregation energies/concentrations when more sites were included.

Figure R4. The atomistic configurations of the W supercells after relaxation. (a) $\Sigma 13$ (230)-GB (208 atoms); (b) $\Sigma 27$ (552)-GB (237 atoms). The atomic sites chosen for the solute substitution are marked in red color while the reference site is marked in blue color.

The electronic descriptors, including Δdip , x_{sp} and the descriptors required by the residual-correction function, are calculated for the defect sites of interest (red atoms in Fig. R4) from the DFT results of the defects in pure metals. The calculated descriptors are then plugged into both the linear+ f_{r-c} and original linear model to predict the interaction energies. The prediction results are validated with the interaction energies from direct DFT calculations. The corresponding parity plots are shown in Figs. R5a and R5b for the W -Re and Ta-Hf systems, respectively.

Figure R5. E_{int}^{fix} predicted from the previous linear model (blue) and the Linear+ f_{r-c} model (orange) in comparison with the results from the direct DFT calculations. The DFT results in the present figures are not included in the dataset for the regression process.

As shown in Figs. R5a and 5b, for both GBs in both systems, the predictions solely from the original linear model already reach fairly good agreements with the DFT results, indicating that the major energy contributions to E_{int}^{fix} can be well captured by the linear model alone. Moreover, by adding the residual-correction function (f_{r-c}), the linear+ f_{r-c} model yields better agreements, especially for the sites where the predictions of the linear model have large deviations. Similar validation results are also observed for the W-Ta, W-Pt and Ta-Os systems, as shown in Fig. S17, respectively. The prediction root-mean-squared errors (RMSE) of the linear+ f_{r-c} model and original linear model are listed in Table R2 below.

Table R2. The root-mean-squared errors (RMSE) of the predictions of the present linear+ f_{r-c} model and original linear model for the testing data set of solute-defect interactions in two complex GBs with 5 matrix-solute element pairs.

	Regression root-mean-squared errors (eV)				
	W-Ta	W-Re	W-Pt	Ta-Hf	Ta-Os
Linear	0.055	0.064	0.317	0.068	0.214
Linear+ f_{r-c}	0.046	0.041	0.148	0.038	0.145

Third, to examine the prediction accuracy and usefulness of the present model, the solute segregation concentrations of the two tested GBs are calculated based on the White-Coghlan (WC) site occupation model^{14,15} with the model-predicted interaction energies, and compared with the those obtained using the DFT-computed interaction energies.

With solute-defect interaction energies at each defect site, we can use the WC model^{14,15} to estimate the GB solute concentration isotherms under an assumption of non-interacting solutes,

$$c_{GB} = \frac{1}{N} \sum_{i=1}^N \frac{1}{1 + \frac{1-c_{bulk}}{c_{bulk}} \exp\left(-\frac{E_{int}^{X,i}}{k_B T}\right)} \quad \text{Eq.R2}$$

where $E_{int}^{X,i}$ is the interaction energy of solute, X , when it occupies the i th of N sites at GB, T is temperature, and c_{bulk} is the solute concentration in the bulk matrix. The GB solute concentration isotherms that calculated using the interaction energies predicted by the linear and linear+ f_{r-c} model for all five solute-matrix pairs are plotted in Figs. R6a and R6b for the $\Sigma 13$ (230)-GB and $\Sigma 27$ (552)-GB, respectively. The isotherms curves obtained with the interaction energies from direct DFT calculations are also included as references. Here, the bulk solute concentration is fixed to be 2 at.%.

As shown in Fig. R6, across a wide temperature range, the interaction energies predicted by the linear+ f_{r-c} model give concentration isotherms that are very close to those obtained with the explicitly DFT-computed interaction energies (dashed and solid lines, respectively). The largest deviation is seen for the case of Pt in W $\Sigma 27(552)$ -GB at high temperature range, which is about 6 at.%. In addition, the curves calculated using the interaction energies solely predicted by the linear model are also in good agreements with the DFT reference, except for the W-Pt system. Further analyses show that such large deviation results from the relatively large error between the DFT result and the linear model prediction for two particular sites near the GB (outliers). It suggests that the linear+ f_{r-c} model can increase the predictability compared with the linear model by reducing the error of certain outlier data due to more complex fitting function and a broader set of electronic structure descriptors beyond Δdip and x_{sp} .

Figure R6. GB solute concentration isotherms calculated based on WC model^{14,15} using the solute-GB interaction energies obtained from direct DFT calculations (solid line) and predicted from the linear (dotted line) and the linear+ f_{r-c} model (dashed line).

In summary, the results in Fig. R6 suggest that, with the present model that established based on the calculations of defects with relatively simple atomistic structures, one may estimate the interaction energies for complex defect structures with reasonably small uncertainty. The predictions would be good enough for approximately describing the properties that are overall determined by the interaction energies of multiple defect sites, such as the isotherms of the solute GB segregation concentration under an assumption of non-interacting solutes. On the other hand, for the defect properties that are only determined by one or two critical sites, the present model may have the risk to yield large errors if the predicted interaction energy of the critical sites happened to be less accurate. Since the residual corrections are established based on non-parametric statistical learning, we expect their accuracy can be further improved in our future research by including more DFT training data.

Finally, the main text and Supplementary Information of the manuscript are carefully revised and reorganized to incorporate these major revisions (Figs. 5 and 6 in the main text and *Sections* 13 and 14 in the Supplementary Information) .

3) Stepping back from the search for universal correlating factors, it is probably important to recognize that many of the quantities studied here can now be computed relatively easily with direct DFT, and with full relaxation. The emergence of high-throughput methods, even if exchanging accuracy for speed by reducing convergence and accuracy of the DFT, has emerged as an approach that can cover a very wide space. The computations of solute interactions with point defects is probably particularly easy, and with grain boundaries is moderately efficient although not for very complex boundaries. Computations of solute/dislocation interaction energies are probably the most challenging due to the system sizes required, but it is also here where the accuracy of the present method may again be insufficient. That is, the interaction energies may be small but the differences between the correlation and the DFT are likely to large enough to be important, while not including relaxations that presumably become more important as the absolute energies become small. The authors might wish to claim that the correlation is meant precisely for issues such as complex grain boundaries that are computational intensive, but they have not established what level of accuracy is really required for physical problems of interest and whether their methods would be useful for alloy design/selection (as noted by one of the reviewers).

Answer: Thanks for the comment and suggestion. We agree with the reviewer that the high-throughput method becomes a powerful tool to screen many materials properties nowadays. Nevertheless, as also pointed by the reviewer, it becomes very difficult or even inapplicable due to the tremendous computational costs if it is used to screen the solute-defect interactions for defect structures with complex geometries, such as the grain boundaries with high index plane and low symmetries or dislocation kinks/jogs, where many DFT calculations have to be performed for many different defect sites in just one complex defect structure (like 5 types of solute elements at $>\sim 50$ defect sites in one supercell with $>\sim 500$ atoms).

Instead of many DFT calculations for different solute elements and different atomic sites surrounding a specific defect, only one DFT calculation for this defect in pure matrix metal is

needed to obtain the local electronic descriptors for our correlation models to map all solute-defect interaction energies. As suggested by the reviewer, additional calculations and analyses were now performed to further validate the prediction accuracy and usefulness of the correlations discovered in the present work. The results of Figs. 5 and 6 in the revised main text suggest that the present model could provide reliable predictions on solute-defect interactions for complex grain boundaries in both Ta and W metals, even though the model itself is established with the calculations of defects with relatively simple atomistic structures. Moreover, due to the localized features of the electronic band in transition metals, in the future it is possible to connect the electronic descriptors with the local atomic structures of the defect by advanced statistical learning methods if more DFT datasets are available for various types of defect configurations, which is our next-step task. As a result, one would be able to directly predict solute-defect interactions with reasonably small uncertainties solely based on the atomic structures of defects in pure metals that obtained by the state-of-art evolutionary structure search algorithms, such as the $\Sigma 27(552)$ -GB tested in the present work. This will be particularly useful for complex defect configurations critical to their kinetics, such as solute interactions with dislocation kinks/jogs and disconnections in grain boundaries, or solute interactions with defect nuclei, which are otherwise hard to directly study using DFT calculations.

4) The authors are to be commended for extensive and detailed replies to the reviewers, extension of their ideas to include sp-d effects in some way and therefore achieve better correlation (albeit with fitting parameters), and for the overall exploration of a promising and potentially very valuable correlation. However, the actual achievement still falls somewhat short of the goals. The proposed universal correlation is not sufficiently universal nor sufficiently accurate. I wish it were otherwise, and that the authors could have achieved their goals. Certainly the field of materials science is filled with rather weaker correlations that have been widely adopted by experimentalists and pursued by modelers due to the ease of computation, but those trends are unfortunate because they are also not accurate and not sufficiently predictive. I believe that the revised paper could/should be published in an archival journal so that the ideas are disseminated and evaluated, and the community will learn from this effort. But publication in Nature Communications is probably not warranted because the limitations of the work make it of much narrower and less definitive value than is usually sought by the editors of Nature Communications.

Answer: Thanks for the reviewer's suggestions. After we revised the manuscript as described above, we think the proposed correlation relation is sufficiently general with acceptable accuracy for the prediction of certain defect properties that are overall determined by the interaction energies of multiple defect sites, like solute segregation near GBs shown above. More importantly, it opens a possible physical + statistical routine has the potential to be used as the screen tools for many first-principles defect studies if more data are generated and investigated not only by the authors but also by the whole computational materials science community. Thus, as suggested by the reviewer, we will share the detailed data on open access repository to speed up the first-principles design of alloys.

References:

1. Lüthi, B., Ventelon, L., Rodney, D. & Willaime, F. Attractive interaction between interstitial solutes and screw dislocations in bcc iron from first principles. *Computational Materials Science* **148**, 21–26 (2018).
2. Wang, J., Janisch, R., Madsen, G. & Drautz, R. First-principles study of carbon segregation in bcc iron symmetrical tilt grain boundaries. *Acta Materialia* **115**, 259–268 (2016).
3. Medvedeva, N. I., Gornostyrev, Y. N. & Freeman, A. J. Electronic origin of solid solution softening in bcc molybdenum alloys. *Physical review letters* **94**, 136402 (2005).
4. Xin, H., Vojvodic, A., Voss, J., Nørskov, J. K. & Abild-Pedersen, F. Effects of d-band shape on the surface reactivity of transition-metal alloys. *Physical Review B* **89**, 115114 (2014).
5. Li, H., Draxl, C., Wurster, S., Pippan, R. & Romaner, L. Impact of d-band filling on the dislocation properties of bcc transition metals: The case of tantalum-tungsten alloys investigated by density-functional theory. *Physical Review B* **95**, 094114 (2017).
6. Al-Zoubi, N. *et al.* Elastic properties of 4d transition metal alloys: Values and trends. *Computational Materials Science* **159**, 273–280 (2019).
7. Zhao, S., Egami, T., Stocks, G. M. & Zhang, Y. Effect of d electrons on defect properties in equiatomic NiCoCr and NiCoFeCr concentrated solid solution alloys. *Physical Review Materials* **2**, 013602 (2018).
8. Pettifor, D. G. *Bonding and structure of molecules and solids*. (Oxford University Press, 1995).
9. Znam, S., Nguyen-Manh, D., Pettifor, D. G. & Vitek, V. Atomistic modelling of TiAl I. Bond-order potentials with environmental dependence. *Philosophical Magazine* **83**, 415–438 (2003).
10. Drautz, R. & Pettifor, D. G. Valence-dependent analytic bond-order potential for transition metals. *Physical Review B* **74**, 174117 (2006).
11. Pettifor, D. G. Theory of energy bands and related properties of 4d transition metals. III. s and d contributions to the equation of state. *Journal of Physics F: Metal Physics* **8**, 219 (1978).
12. Sutton, A. P. *Electronic structure of materials*. (Clarendon Press, 1993).
13. Hammer, B. & Nørskov, J. K. Theoretical surface science and catalysis—calculations and concepts. *Advances in Catalysis* **45**, 71–129 (2000).
14. Huber, L., Hadian, R., Grabowski, B. & Neugebauer, J. A machine learning approach to model solute grain boundary segregation. *npj Computational Materials* **4**, 64 (2018).
15. White, C. L. & Coghlan, W. A. The spectrum of binding energies approach to grain boundary segregation. *Metallurgical Transactions A* **8**, 1403–1412 (1977).

16. Loader, C. *Local regression and likelihood*. (Springer Science & Business Media, 2006).
17. Stone, C. J. Consistent nonparametric regression. *The annals of statistics* **5**, 595–620 (1977).
18. Cleveland, W. S. Robust locally weighted regression and smoothing scatterplots. *Journal of the American statistical association* **74**, 829–836 (1979).
19. De Jong, M. *et al.* A statistical learning framework for materials science: application to elastic moduli of k-nary inorganic polycrystalline compounds. *Scientific Reports* **6**, 34256 (2016).
20. Frolov, T. *et al.* Grain boundary phases in bcc metals. *Nanoscale* **10**, 8253–8268 (2018).
21. Zhu, Q., Samanta, A., Li, B., Rudd, R. E. & Frolov, T. Predicting phase behavior of grain boundaries with evolutionary search and machine learning. *Nature Communications* **9**, 467 (2018).

REVIEWERS' COMMENTS:

Reviewer #3 (Remarks to the Author):

The authors have done extensive and much appreciated work in their manuscript revision, and it is a clearly interesting study. However, the impact, as have been argued in previous reviews, is not in my opinion up to the standards of Nature Communications. The work certainly merits publication and I would suggest Physical Review Materials as a suitable journal.

I will argue the case briefly: The original claims and manuscript title were of a level for Nat.Comm., but, through the review process, the authors have correctly downplayed the "universal" character of the method to a proper and correct level in terms of conclusions and claims. Now, the method still is not sufficiently accurate to replace direct DFT calculations for the truly sensitive interactions. There is a lot of data that differ by magnitudes that are physically important. One very crucial aspect is to provide accurate predictions in the proximity of no interaction, i.e. crossing 0. Also, attractive interactions are also more fundamentally more important than repulsive are. The lack of "near DFT-level" predictability for these regions is the main reason I fear this is not going to have the suggested impact, and the impact Nature Communications is searching for.

I will, however, iterate that the work merits publication and after this extensive review process should have a much easier time going through e.g. Phys.Rev.Mater.

Reviewer #4 (Remarks to the Author):

The authors have revised the work significantly, adding additional computations, analysis, and results. The authors are headed down a potentially very valuable path, and set out an approach that may be expandable and a guide for considerable future work by others in the field (using the details features of the band structure to understand solute energies in defect structures). I believe the paper can be accepted for publication in its present form.

REVIEWERS' COMMENTS:

Reviewer #3 (Remarks to the Author):

The authors have done extensive and much appreciated work in their manuscript revision, and it is a clearly interesting study. However, the impact, as have been argued in previous reviews, is not in my opinion up to the standards of Nature Communications. The work certainly merits publication and I would suggest Physical Review Materials as a suitable journal.

I will argue the case briefly: The original claims and manuscript title were of a level for Nat.Comm., but, through the review process, the authors have correctly downplayed the "universal" character of the method to a proper and correct level in terms of conclusions and claims. Now, the method still is not sufficiently accurate to replace direct DFT calculations for the truly sensitive interactions. There is a lot of data that differ by magnitudes that are physically important. One very crucial aspect is to provide accurate predictions in the proximity of no interaction, i.e. crossing 0. Also, attractive interactions are also more fundamentally more important than repulsive are. The lack of "near DFT-level" predictability for these regions is the main reason I fear this is not going to have the suggested impact, and the impact Nature Communications is searching for.

I will, however, iterate that the work merits publication and after this extensive review process should have a much easier time going through e.g. Phys.Rev.Mater.

Answer: Thanks for the reviewer's appreciation for our efforts and all insightful comments, which improved our manuscript significantly. However, we have emphasized the broad impacts of this manuscript are not to "replace direct DFT calculations for the truly sensitive interactions" right now, but the following two aspects. First, as indicated by results in Figs. 5 and 6, our method provides a fast-screening tool to efficiently predict the defect properties that depend on the statistical **average effects** of defect-solute interactions, such as solute-segregation isotherms. Second, as pointed out by the Reviewer #4, our studies "set out an approach that may be **expandable** and a guide for considerable future work by others in the field (using the details features of the band structure to understand solute energies in defect structures)." In our opinion, the "future work by others" may include machine-learning-based methods for alloy design, which are under fast development and highly depend on the constructions/selections of descriptors.

Meanwhile, for the cases of "the truly sensitive interactions", we acknowledge that the present model is not always accurate to predict the individual weak defect-solute interaction at a particular defect site in the limit of no interaction (0.01~0.05 eV), but this issue can be further improved if 1) more training data are included, 2) more representative and robust descriptors of electronic/atomistic structures are discovered, and 3) more advanced regression methods are applied. To achieve this goal, we have uploaded all of our original raw data (DFT inputs and key outputs) and codes to an open-access database to invite more people in computational materials science community to explore this direction. We have added these discussions in the manuscript.

In addition, the reviewer mentioned "attractive interactions are also more fundamentally more important than repulsive are.". We think there could be some misunderstanding. Most of the solute-defect interactions studied in this manuscript are attractive type as shown in Figs. 1~5 ($E_{\text{int}}^{\text{fix}} > 0$ means attractive interactions in our manuscript), and our regression/predication results indeed show

good accuracy for these attractive interactions. We also add clear explanation in the revised manuscript to avoid any misunderstanding for future readers.

Reviewer #4 (Remarks to the Author):

The authors have revised the work significantly, adding additional computations, analysis, and results. The authors are headed down a potentially very valuable path, and set out an approach that may be expandable and a guide for considerable future work by others in the field (using the details features of the band structure to understand solute energies in defect structures). I believe the paper can be accepted for publication in its present form.

Answer: Thanks for the reviewer's appreciation for our efforts and all insightful comments, which improved our manuscript significantly.